# Levels of Variation in Subordinates of Immediate Succession in Current Spanish

Avel·lina Suñer Gratacós 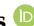

Departament de Filologia i Comunicació, Universitat de Girona, 17071 Girona, Spain; avellina.sunyer@udg.edu

**Abstract:** In this paper, I analyze, from a compositional perspective, the relevant features to construct the interpretation of immediate succession between a subordinate event and the event that takes place in the main sentence. Among all the components involved in the construction of the meaning of immediate succession, I focus particularly on the subordinators, which present a mosaic of variation in current Spanish. The key ideas that can be derived from the data analysis are the following. First: subordinators of immediate succession are the *loci* of variation of temporal subordinates. Second: a subordinator of immediate succession is a "linguistic variable" that can be syntactically materialized in different forms by applying general rules that do not change the meaning, although sometimes they do change the grammatical category. Third: in the diachronic evolution of Spanish, several patterns of internal structure have emerged for immediate succession subordinators. However, most of them have ceased to be productive, although some subordinators that were coined with these patterns have survived as fossils in the current language. Fourth: the only productive pattern in the present language can be reduced to the Adv (immediacy) + *que* scheme, which goes back to Late Latin.

**Keywords:** immediate succession; subordination; grammaticalization; compositionality; social and geographic variation; linguistic variable; preposition; conjunction and complex conjunction

## 1. Introduction

The aim of this article is to provide an overview of the geographical variation of subordinators of immediate succession in current Spanish. However, reference is made to diachronic and social variation when necessary to explain some aspects of geographic variation. As indicated by Eberenz (1982), Espinosa (2010) and Herrero (2018), subordinators that express this notion manifest the highest degree of variation within temporal subordinates. This paper focuses precisely on the variation manifested by these subordinators to elucidate which forms may be regarded as variants of the same subordinator and which patterns of formation of their internal structure are still productive in the current language.

### 1.1. Anteriority, Posteriority, and Immediate Succession

Temporal subordinates express a specific temporal relationship between a subordinate event (or state of affairs) and the event (or state of affairs) taking place in the main sentence. Between the two events, there may be a relationship of simultaneity (total or partial) or of succession in time. Two events are simultaneous if they coincide (totally or partially) on the time axis. If one of them follows the other, the relationship may be either of anteriority or posteriority depending on which event is taken as the point of reference. The Spanish grammatical tradition takes the main sentence as the point of reference for assigning the values of anteriority or posteriority (e.g., RAE 1973, III, § 3.21; Eberenz 1982, pp. 295–98; Méndez 1995, II. § 2; Herrero 2005, § 6.6.3; RAE-ASALE 2009, § 24.5k; Eberenz 2014, § 34.1). Following this criterion, in (1a), the temporal subordinate introduced by *después de que* 'after' expresses posteriority since the main event "your mother phoned" occurs after that of the subordinate clause "you left". Similarly, the subordinate clause headed by *antes de*

*que 'before'* in (1b) indicates anteriority since the main event occurs chronologically before that in the subordinate clause.

(1)  a.  [Después de que te fueras], tu madre telefoneó.
         'After you left, your mother called.'
     b.  [Antes de que tu madre telefoneara], te fuiste.
         'Before your mother called, you were gone.'

On the contrary, in key studies on temporal subordination (e.g., Heinamaki 1978; Kortmann 1997, among others) it is the subordinate clause that has been taken as a reference to determine the values of anteriority and posteriority. From this point of view, the subordinate clause of (1a) expresses anteriority, since the event takes place in a time segment before that of the main clause, while the subordinate clause of (1b) expresses posteriority since it expresses an event that happens after that in the main clause on the time axis.

Regardless of whether the main clause or the subordinate clause is taken as the point of reference, the relation of immediate succession occurs when two events (or states of affairs) chronologically ordered on the time axis are separated by an interval or segment of time that must necessarily be short.

In this article, I will take, as a point of reference, the interval between the two events (or states of affairs) that make up the complex period. This approach emphasizes the significance of the fact that the previous event (or state of affairs) has culminated precisely at the beginning of the short time interval and that the end point of the interval coincides with the beginning of the second event. It does not seem to be a coincidence that immediate succession is a temporal pattern that is consistent with the way events occur in extralinguistic reality and, by extension, with relevant logical–pragmatic relations such as causality that can be established between two successive events; see Herrero (2018).

This approach is consistent with the generally accepted idea that temporal subordinators are two-place predicates, as convincingly shown by Heinamaki (1978), Hitzeman (1991), García Fernández (2000), Dermidache and Uribe-Etxebarría (2004), Bosque and Bravo (2015), and others. According to this idea, the subordinator of immediate succession *en cuanto* 'as soon as' in (2) selects the subordinate event as its internal argument (IA) while the main event is the external argument (EA) selected by the predicate (subordinator + subordinate event).

(2)  [$_{EA}$ Los precios de los cereales subieron] [*en cuanto* [$_{IA}$ empezó la guerra de Ucrania]]
     'Grain prices went up as soon as the Ukrainian war started.'

One of the benefits of this analysis is that it provides an elegant way to explain the temporal correlation between the verb of the subordinate clause and the verb of the main clause as a selection requirement of the temporal predicate of immediate succession *en cuanto* 'as soon as' (e.g., Carrasco 1998, chap. 8; García Fernández 2000, chap. 10).

A Spanish speaker does not necessarily have to use a specific subordinator of immediate succession to express this meaning. In fact, it is possible to indicate that one event immediately follows another through productive syntactic rules, e.g., by adding a differential indicating a short period of time, *poco* 'shortly', or by attaching a modifier expressing immediacy such as *inmediatamente* 'immediately' or an approximative adverb *justo* 'just' (cf. RAE–ASALE 2009, § 29. 3m–n) to a comparative subordinator of simple posteriority (*después de que* 'after') or anteriority (*antes de que* 'before'), as the examples in (3) show.

(3)  a.  [Poco ~ Inmediatamente ~ Justo después de que te fueras], tu madre telefoneó.
         'Shortly ~ Immediately ~ Shortly after you left, your mother called.'
     b.  [Poco ~ Inmediatamente ~ Justo antes de que tu madre telefoneara], te fuiste.
         'Shortly ~ Immediately ~ Just before your mother called, you were gone.'

However, the procedure illustrated in (3) is unusual in contemporary Spanish since the speaker prefers to resort to a large repertoire of subordinators that grammatically encode the notion of immediate succession in their internal structure.

*1.2. The Compositional Perspective: Building Up the Meaning of Immediate Succession*

In recent reference grammars (e.g., Huddleston and Pullum 2002; Maiden and Robustelli 2007; RAE-ASALE 2009; IEC 2016), there has been no specific section devoted to temporal subordinates. The information is broken down into several subsections on adjuncts, Prepositional Phrases (PPs), Adverbs (Advs), Conjunctions (Conjs), and relative subordinates, among others. This system of organizing the description of temporal subordinates, and, by extension, those expressing immediate succession, is the logical consequence of the assumption that the temporal content is not embodied exclusively in the temporal subordinator but is divided into different relevant components of the sentence. In other words, the meaning of immediate succession between two events is constructed compositionally by means of the subordinator heading the temporal subordinate, together with the tense, the mood, the syntactic aspect of the verbs involved, their lexical aspect (*Aktionsart*), and, optionally, the presence of temporal and aspectual modifiers, quantification, negation, and the relative order between the main sentence and the subordinate (see Haegeman 2012, chap. 5; Brucart and Gallego [2009] 2016).

Subordinators of immediate succession are the result of the grammaticalization[1] of heterogeneous syntactic categories. Some of them are PPs such as *deque* 'as soon as' (rural Spanish) and *desde que* 'as soon as' (Western Andalusian, Canarian, Dominican, and other Caribbean varieties). The most common are AdvPs or adverbial expressions followed by the conjunction *que* (*luego que*, *immediatamente que* 'as soon as') or without it (*apenas* 'as soon as', *no bien* 'as soon as'). There are also subordinators of immediate succession coming from modal structures (*así que* 'as soon as') or comparative QPs (*tan pronto como*, 'as soon as', *tan pronto* 'as soon as'). Finally, some of these subordinators are derived from an NP or QP followed by the conjunction *que* 'that': *una vez que* 'once' 'as soon as' and *lo que* 'as soon as' (North Patagonia).

Some recent approaches to adverbial subordination question whether the different syntactic strategies used in this type of subordination constitute a specific formal class (e.g., Bosque and Gutiérrez-Rexach 2009, § 11.9; Brucart and Gallego [2009] 2016; RAE-ASALE 2009, § 1.13p-u; Haegeman 2012, § 5.1). In fact, these authors claim that adverbial subordinates could be assimilated to completive or relative subordinates depending on the syntactic operations needed to express their dependence on their main clause. If this premise is correct, adverbial subordinates, and, by extension, those of immediate succession, are derived from the basic structure of (4), in which the most prominent node, P (or an equivalent category), which provides the adverbial/adjunct value, selects, as its internal argument, a Complementizer Phrase; see Brucart and Gallego ([2009] 2016), and Haegeman (2012, § 5.1).

(4)     [$_{PP}$ P(reposition). . . [$_{CP}$ C(omplementizer) . . .]]

In (4), P must be understood in an abstract way since it can be lexically manifested through different syntactic categories that have, in common, the fact of expressing an adverbial meaning such as P, in (5a), and Adverb (Adv), as illustrated in (5b).

(5)   a.   [$_{PP}$ [$_P$ De] [$_{CP}$ [$_C$ que] [$_{IP}$ acabe de comer]]], me acostaré en siesta. (A.M., young
                  woman, 2020, Ceclavín, Cáceres, Spain)
                  'As soon as I finish eating, I'll take a nap.'
       b.   La *seasonal depression* entró a mi cuerpo [$_{ADVP}$ [$_{ADV}$ inmediatamente] [$_{CP}$ [$_{CP}$ que] [$_{IP}$
                  empezó a oscurecer antes de las 5pm]]]. (*Twitter*: young woman, 7 November 2023,
                  Sonora, Mexico]
                  'The seasonal affective disorder hit my body as soon as it got dark before 5 p.m.'

P, in (4), may also emerge as a complex conjunction such as *enseguida de que* 'as soon as' in (6).

(6)      Hay que decir que la inspección la hicieron [$_{ADVP}$ [$_{ADV}$ enseguida] [$_{PP}$ [$_{P}$ de] [$_{CP}$ [$_{C}$ que] [$_{IP}$ puse el reclamo el 8 de octubre]]]]. (*Twitter*: young woman, 20 October 2023, Quito, Ecuador)
'It must be said that the inspection was done as soon as I filed the claim on October 8th'.

A complex conjunction, such *enseguida de que* 'as soon as' in (6), is a linguistic expression made up of one or several words generally followed by the Conj *que* 'that'. Each of these words has, in isolation, an autonomous use, but through a diachronic process of grammaticalization, these words become an indivisible unit functionally equivalent to a simple Conj. Consequently, the members of a complex conjunction cannot be replaced by synonyms or disaggregated by means of the application of any syntactic rule.

The subordinator can also be expressed by an adverb such as *apenas* 'as soon as', in (7a), which introduces the finite subordinate clause without the conjunction *que* 'that'. Some authors, such as Brucart and Gallego ([2009] 2016), suggest that in (7a), *apenas* 'as soon as' behaves similarly to the temporal relative *cuando* 'when', in (7b), since a single lexical item agglutinates the adverbial value and that of the complementizer que 'that'.

(7)    a.    [Apenas amaneció], Juan salió de la recámara y fue a sentarse en el patio. (CORPES XXI: Homero Aridgis, *La zona del silencio*, 2005, México)
'As soon as dawn broke, Juan left the bedroom and went to sit on the patio.'
        b.    [Cuando amaneció] nos quedamos cerca de la ciudad de San Miguel. (CORPES XXI: Lurgio Gavilán, *Memorias de un soldado desconocido,* 2017, Perú]
'When dawn broke, we were near the town of San Miguel.'

There are many temporal subordinators that come from the grammaticalization of adjuncts or modifiers of the predicate that do not merge with the conjunction *que* 'that': *mientras* 'while', *apenas* 'as soon as', *no bien* 'as soon as', and *conforme* 'as', among many others; see Eberenz (1982, p. 293), Pavón (2012, § 4.1.2), and RAE-ASALE (2009, § 31.13). However, they are not fully equivalent to relatives because they lack some of their properties such as having an antecedent and being able to form emphatic relative clauses, as in (8); see RAE-ASALE (2009, § 40.10–11).

(8)      Recuerdas que yo tenía un catus (*sic*) que se llamaba Shelby? Ernesto, pues [cuando murió fue cuando le deje de contar chisme (sic)] (*Twitter*: young man, 7 July 2023, Bogotá, Colombia)
'Do you remember that I had a cactus named Shelby? Ernesto, well, when he died, that's when I stopped telling him gossip.'

The subordinator *una vez que* 'as soon as' 'once', in (9), does not apparently fit into the general scheme in (4).

(9)      La gente necesita entender que [una vez que pasas por tu peor momento solo], ya no te importa quién se queda en tu vida. (*Twitter*: young man, 11 November 2023, Medellín, Colombia)
'People must understand that [once you go through your worst time alone] you no longer care who stays in your life.'

The key question is how the initial P in rule (4) is formally manifested in *una vez que* 'once' 'as soon as', in which this element is not identifiable a priori. This problem must be related to the fact that in Spanish, temporal adjuncts can be NPs, such as *el sábado* 'on Saturday' in (10).

(10)     El sábado llegaron 86.400 dosis de lotes de vacunas. (CORPES XXI: www.latribuna.hm (accessed on 1 September 2023), 25 May 2021, Tegucigalpa, Honduras)
'On Saturday, 86,400 doses of vaccine batches arrived.'

The way in which *el sábado* 'on Saturday' in (10) obtains an abstract case mark without the contribution of an overt P is a controversial question. Larson (1985) and Tremblay (1991) argue that temporal NPs are in fact covert PPs because P is not lexically expressed but

is active from an interpretive point of view. On the other hand, Carrasco (1998), García Fernández (1999, § 48.1.2.1; 2000, V, § 2.2), and Fábregas (2020) suggest that temporal NPs are not adjuncts (and therefore they need no P to assign Case) but arguments of a functional projection, usually TP.

In current Spanish, there are no subordinators that come from temporal NPs. In fact, all of them are temporal QPs: *una vez (que)* 'as soon as' 'once', *cada (vez) que* 'every time that', and *lo que* 'as soon as' (North Patagonia). As argued in Kayne (2014) for *once* (English), the presence of Q (*on < once*) plays a determining role to trigger a grammaticalization process through which the adjunct is reanalyzed as a functional element equivalent to Conj.

In addition to the meaning grammatically encoded in the subordinator, the interpretation of immediate succession between two events requires that the verbs involved in the two clauses follow a specific pattern of temporal correlation.

The most common verbal forms in subordinates of immediate succession in contemporary Spanish are the past simple, in (11a), and the present perfect, in (11b), which have the function of highlighting, in a complex period, the relevance of the culmination of the first event with respect to the second, which begins after a short time interval. The use of the simple past, in (11a), or the present perfect, in (11b), in the subordinate clause makes it possible to refer to two retrospective events that have occurred in the past and are therefore factive.

(11)    a.    [En cuanto salió-IND.PASTS.3SG del quirófano] le rodearon decenas de policías. (CORPES XXI: Guillermo Arriaga, *Salvar el fuego*, 2020, México)
            'As soon as he came out of surgery, he was surrounded by dozens of police officers.'

        b.    Y [en cuanto ha+salido-IND.PRES.PF.3SG por la puerta] me han entrado muchas ganas de llorar. (CORPES XXI: Eloy Moreno, *Invisible*, 2018, Spain)
            'And as soon as she walked out the door I felt like crying.'

Although the present subjunctive, in (12a), and the present indicative, in (12b), have no perfective value, they can also appear in this type of subordinate.

(12)    a.    [En cuanto salga-SUBJ.PRES.1SG del bache], nos vamos a dar un homenaje que vas a llorar. (CORPES XXI: Daniel Guzmán, *A cambio de nada*, 2015, Spain)
            'As soon as I get out of trouble, we're going to have so much fun that you'll be thrilled.'

        b.    Pero [en cuanto salen-IND.PRES.3PL del gobierno], culpan a los socialistas de sus propios errores de gestión. (CORPES XXI: José Carlos Díez, www.elpais.com (accessed on 28 July 2023), 2 november 2018, Spain)
            'But as soon as they leave the government, they blame the socialists for their own management mistakes.'

The present subjunctive places the two events in the future, so it gives rise to a non-factive prospective interpretation. If the subordinate of immediate succession includes an indicative present, in (12b), the interpretation of the whole complex period can be iterative. Regardless of the verb tense used in the subordinate clause of immediate succession, the verb of the main clause must place the beginning of the second event after a short interval following the culmination of the first event.

More marginally, the subordinate clause can contain a form of the past imperfect, in (13a), and, to a lesser extent, a past perfect ("pretérito anterior" in Spanish), in (13b).

(13)    a.    […] [en cuanto venía-IND.PAST.IMP.3SG el verano], ya no, no podíamos ir a la escuela (COSER-0939_01, old woman, 27 March 1993, Quintana de los Prados, Espinosa, Burgos, Spain)
            'As soon as summer came, we could no longer go to school.'

        b.    [En cuanto hubo+salido-IND.PAST.PF.3SG del bar] Lizarso lo recriminó. (CORPES XXI: Dolores Redondo, *Esperando al diluvio*, 2022, Spain)
            'As soon as he had left the bar, Lizarso gave him a rebuke.'

The imperfective past, like the present tense, does not delimit the end point of the first event. However, in (13a) it is interpreted in a special way, called the *historical or narrative imperfect* (see Eberenz 1982, pp. 317–19; 2014, § 34.5.1.4; García Fernández 2000, p. 248; RAE-ASALE 2009, § 23.12o and § 24.5o-t; Espinosa 2010; Pavón 2013), which is common in texts that narrate events that happened in the past and are repeated cyclically (as is the case with the rural Spanish texts in the COSER corpus). In (13b), the "pretérito anterior" verbal form is constructed by adding a perfective form of the auxiliary *haber* 'to have' and the past participle *salido* 'gone out', thus referring to an event that culminated in the past. Nevertheless, it has an archaic flavor, since its use declined from preclassical and classical Spanish, and it is preserved only in written texts; see Rodríguez Molina and Octavio de Toledo (2008).

Finally, some subordinators, such as *nada más* 'as soon as', are combined with an infinitive verb (INF) that does not express temporal features by itself. In these cases, the interpretation of immediate succession depends on the subordinator and the verb tense of the main clause, which must express an event immediately following the short interval after the first event, as in (14).

(14)  [Nada más operarse-INF de la fractura de la cadera] la abuela Concha tuvo que empezar a dar pequeños pasitos con las muletas y el andador para evitar posibles complicaciones. (CORPES XXI: David Martínez Álvarez, *El acercamiento de la mujer cactus y el hombre globo*, 2023, Spain)
'As soon as she underwent surgery for her hip fracture, Grandma Concha had to start taking small steps with crutches and a walker to avoid possible complications.'

So far, it has become clear that the subordinator and the temporal correlation between the verbs involved in the complex period play decisive roles in the process of constructing the meaning of immediate succession. In the following paragraphs, we will analyze whether the linear order between the main and subordinate clauses is another relevant component in this process.

As the pair of examples in (15) show, some subordinates of immediate succession can both precede and follow the main clause.

(15)  a.  [En cuanto entró-IND.PASTP.3SG], la mujer se sentó y sacó la bolsa de dinero. (CORPES XXI: Xabier Gutiérrez, *El aroma del crimen*, 2015, Spain)
'As soon as she entered, the woman sat down and took out the bag of money.
b.  La saludé [en cuanto entró-IND.PASTP.3SG]. (L.V., old man, 17 May 2023, Veganzones, Segovia, Spain
'I greeted her as soon as she came.'

The choice of the linear order between a temporal subordinate and its main clause is not optional but is determined by several factors. Diessel (2001, 2008) argues that iconicity plays a crucial role in determining the positions of temporal subordinates with respect to their main clauses in spoken and written English. Although temporal subordinates show a strong preference for following their main clauses, the linear order in which the two sentences are arranged reflects the chronological order in which the two events occurred. In short, temporal subordinates expressing an earlier event tend to precede the main clause more often than temporal subordinates expressing later events. Guerrero et al. (2017) came to similar conclusions for Spanish. Despite the conclusions about the linear order in which temporal subordinates of anteriority and posteriority are arranged in contemporary Spanish, some studies on the diachronic evolution of subordinators of immediate succession, such as those by Eberenz (1982), Méndez (1995, IV, § 2), and Herrero (2005, § 6.3.3.2.2), have emphasized that most of the subordinates expressing this notion in Old Spanish tended to precede the main clauses, and in some cases, this was the only possible option. Although I have not conducted a quantitative study of the preferred linear order for subordinates of immediate succession in the current language, my general impression is that some of them prefer the anteposition to their main clause. This view, which should be confirmed by quantitative studies, seems to indicate that immediate succession does not conform to

relations of anteriority or posteriority (see Diessel 2001, 2008), but rather emphasizes the relevance of the short interval between the two successive events.

*1.3. Historical Background: The Forging of New Subordinators of Immediate Succession*

All the evolutionary stages that lead from Latin to present-day Spanish show a constant renewal, change, and the forging of new subordinates of immediate succession. Indeed, none of the subordinators of immediate succession in Classical Latin were preserved in Medieval Spanish, nor in later stages of the language (see Herman 1963, pp. 88, 101; Herrero 2018). In contrast, subordinators expressing basic temporal notions such as anteriority (*antes de que* 'before' < Lat. ANTEQUAM) or posteriority (*después de que* 'after' < Lat. POSTQUAM) are stable over time, and they do not give rise to much diachronic, geographic, and social variation; see Pinkster (2015, pp. 609–11, 638–39; 2021, pp. 258–59).

In Early and Classical Latin, immediate succession was expressed either by the connector SIMUL 'as soon as', alone or combined with the coordinating conjunctions AC, ATQUE, or ET 'and' (see Pinkster 2021, 16.22), or by the subordinators CUM PRIMUM (see Pinkster 2021, II.258), MOX UBI, and MOX UT 'as soon as' (see Herrero 2018; Pinkster 2021, II.258, II.264). New subordinators such as MOX CUM 'as soon as' and MOX QUOD 'as soon as' appeared in Late Latin (see Herman 1963, pp. 88, 101; Herrero 2018). Their internal structure consisted of an adverb expressing immediacy, MOX 'soon', to which a generic subordinator such as CUM 'when' or QUOD 'that' was attached.

Medieval Castilian did not preserve the subordinators of immediate succession from previous evolutionary stages, although it did retain a syntactic pattern inherited from Late Latin, in (16), which is still productive in the current language, cf. Section 3.2.2.

(16)    SUBORDINATION PATTERN: Adv (immediacy) + *que*

In addition to the introduction, this article contains three other sections. In the second one, I indicate the provenance of the data and the methodological precautions I took in collecting them. The third section contains two subsections describing the different levels of variation of subordinators of immediate succession in Spanish. In the first one, the notion of the "linguistic variable" is applied to subordinators that can give rise to a constellation of variants with the same meaning. These variants are built around a common lexical or grammatical element, which generally provides the component of immediacy, but differ in the functional architecture associated to them. The second subsection identifies the different patterns of the internal structure of subordinators of immediate succession and determines which of them are still productive in the current language. The main ideas and contributions of this article are summarized in Section 4, where some of the problems encountered in analyzing the data are discussed.

## 2. Data

The data analyzed in this paper have different origins depending on the phenomenon to be illustrated. Those of the current language come from six main sources which, in my opinion, offer a very faithful picture of the linguistic reality.

Firstly, I extracted examples from the social network *X* (formerly known as *Twitter*), which provides access to written texts that are not very planned and with a high degree of orality, making it possible to identify variants that have not yet penetrated the written language. In all cases, the *X/Twitter* data were checked to ensure that they illustrated a statistically relevant linguistic phenomenon before being included in the paper. Specifically, for a phenomenon documented on *X/Twitter* to be considered worthy of mention in this article, there had to be at least 15 examples with the same geolocation.

Secondly, I also extracted data from *CORPES XXI* (*Corpus of 21st Century Spanish*, Real Academia Española, https://www.rae.es/banco-de-datos/corpes-xxi accessed from 1 June 2022 to 14 November 2023), which contains geolocalized examples of 21st-century Spanish.

In the third place, I gleaned rural Spanish data from *COSER* (*Oral and Sonorous Corpus of Rural Spanish*, Inés Fernández Ordóñez dir., http://www.corpusrural.es/ accessed from

1 June 2022 to 14 November 2023), which collects oral texts from elderly speakers in rural areas of Peninsular Spanish between 1990 and 2017; see Fernández Ordóñez and Pato (2020).

Fourth, other data were taken from *PRESSEA* (*Corpus of the Project for the Sociolinguistic Study of Spanish in Spain and America*, https://preseea.uah.es/ accessed from 1 June 2022 to 14 November 2023), which collects recordings classified according to the degree of formality from different Spanish-speaking cities between 1990 and the present; see Moreno Fernández and Cestero Mancera (2020).

Fifth, some data were extracted from Davies' *Corpus del Español WEBDIALECTS* (https://www.corpusdelespanol.org/web-dial/ accessed from 1 June 2022 to 14 November 2023), which contains written texts on the Web classified according to their geographical origins.

Finally, in the sixth place, some examples were taken from interviews conducted by the author with speakers of different varieties of Spanish to clarify whether my interpretation of the data was correct. Unfortunately, I was not able to obtain informants for all the varieties of Spanish discussed in the article. In these cases, I relied on the analyses provided in the bibliographic references. The initials of the interviewee's name, his or her origin, and the date on which the information was obtained appear next to each example. All interviewees agreed to have their data used for this article.

Additionally, I have searched for data in *CREA* (*Reference Corpus of Current Spanish*, 1975–2002, Real Academia Española, https://www.rae.es/banco-de-datos/crea accessed from 1 June 2022 to 14 November 2023) to see whether a newly formed subordinator could be documented from the end of the 20th century.

Finally, I have occasionally used *CORDE* (*Diachronic Corpus of Spanish*, origins—1975, Real Academia Española, https://www.rae.es/banco-de-datos/corde accessed from 1 June 2022 to 14 November 2023) and *CORDIAM* (*Diachronic Corpus of American Spanish*, 1492–1900, Concepción Company dir., https://www.cordiam.org/ accessed from 1 June 2022 to 14 November 2023) to document the origin of a given subordinator. The source of each example is given next to it.

Despite all the methodological precautions that have been taken, the reader should bear in mind that the aim of this chapter is to describe the main trends in the geographical and social variation of the subordinators of immediate succession in contemporary Spanish. In this sense, the results obtained are partial since they should be confirmed (or, if necessary, refuted) by more in-depth field studies that will allow more reliable conclusions to be drawn using quantitative methods.

## 3. Results: Levels of Variation of Subordinators of Immediate Succession

As has happened in the historical process of evolution of the subordinators of immediate succession (see Section 1.3), the mosaic of variation that these items currently display is not stable since, over the last decades, some subordinators have fallen into disuse (*así que* 'as soon as'; *luego que* 'as soon as', which is preserved only in American Spanish and is no longer used in European Spanish); see Carabedo (2009), Eberenz (2014, p. 4246), and Herrero (2018), while other new ones have recently appeared in the oral registers (*al nomás que* 'as soon as')

Herrero (2018) argues that the speakers' constant need to renew these subordinators is related to the relevance of the notion of immediate succession that they express, which makes it possible to emphasize that the second event occurs as a result of the first one.

In the variation shown by the subordinators of immediate succession, a distinction must be made among (a) changes affecting the same subordinator, which generally result in an extension of the contexts in which it can appear (see Section 3.1); (b) the creation of new subordinators (*al nomás que* 'as soon as'); and (c) the fall into disuse of some of them (*así que* 'as soon as'; *tan luego como* 'as soon as'); see Eberenz (2014, p. 4246) and Herrero (2018).

This section is divided into two subsections. The first one deals with the variants associated to a given subordinator. In the second, I describe the main patterns of formation of the internal structures of the subordinators of immediate succession in current Spanish.

### 3.1. "Variants" of the Same Subordinator

Some subordinators of immediate succession can give rise to different variants that have the same meaning but with different functional architectures. A speaker may have access to several of these variants (let us call it a *Pool of variants*, as in Adger's (2007, 2014) terminology), but the choice of one of them is not free but organized according to contextual factors. The subordinator chosen depends, among other things, on the notion that a given variant is more lexically accessible than another because it has been routinized, as well as the geographical origin and age of the speaker, his cultural level, or the appropriateness considering the register used by his interlocutors. For this type of variation, I use the notion of the "linguistic variable" developed by Labov (1966, 1978). A subordinator is, then, a linguistic variable X that gives rise to different variants with the same meaning ($x_1$, $x_2$, $x_3$,... $x_N$), depending on contextual factors; see Adger (2007, 2014) and subsequent work. I illustrate this idea with an example.

According to Herrero (2018), *en cuanto* 'as soon as' is the most common subordinator in both spoken and written European Spanish to express immediate succession, as in (17). *En cuanto* 'as soon as' is therefore one of the possible variants of the abstract subordinator EN CUANTO, which can also be expressed by other variants such as *cuanto que* 'as soon as', in (18), and *en cuanto que* 'as soon as', in (18), both from rural European Spanish. From a compositional point of view, all these variants are built from the quantifier *cuanto* 'how much', which, in this case, expresses a short period of time corresponding to the interval between the two successive events[2].

(17)    [En cuanto veo-IND.PRES.1SG que me pongo afónica], no me hago ni caso. (CORPES XXI: Nena Daconte, *Tenía tanto que darte*, 2022, Spain)
        'As soon as I see that I am getting hoarse, I don't pay any attention to myself.'

(18)    Aquí to las noches iban a la ronda. [Cuanto que una noche no..., no iba-IND.PASTIMP.3SG], la riña al otro día (COSER-1020_01, old man, 9 March 1991, Talabán, Cáceres, Spain)
        'Here, every night they went to the round, as soon as one night they didn't..., they didn't go, there was a fight the next day.'

(19)    [En cuanto que hace-IND.PRES.3SG frío], la gallina deja de comer. (COSER-2910_01, old woman, 25 November 1995, Manzanares el Real, Madrid, Spain)
        'As soon as it gets cold, the hen stops eating.'

A speaker of rural European Spanish may choose the variant *en cuanto*, in (17), in the written language but opt for the variants *cuanto que* 'as soon as', in (18), or *en cuanto que* 'as soon as', in (19), if he is talking to people from his village. Although the meaning is the same, the syntactic category to which each variant belongs is different: *en cuanto* 'as soon as' has an internal structure comparable to *apenas* 'as soon as', since it does not merge with the conjunction *que* 'that', whereas *cuanto que* 'as soon as', in (18), and *en cuanto que* 'as soon as', in (19), are complex conjunctions; see Section 1.2.

### 3.2. Patterns of Formation of the Internal Structure of Subordinators of Immediate Succession

The main objective of this section is to describe the syntactic patterns followed by the internal structures of subordinators that head subordinates of immediate succession in present-day Spanish. The primary interest of the description is to determine which of these patterns are still productive in the current language to forge new subordinators and which of them have been abandoned along the evolutionary process. The second objective is to determine which component provides the value of immediacy in each of the subordinators analyzed.

In the following subsections, it will become clear that isolating the different internal structure patterns of subordinators of immediate succession is a complex task since many of them have variants that hybridize with other formation patterns, other have changed their meanings during their diachronic itinerary, and, finally, some subordinators of the current language are fossilized remains of an internal structure pattern that ceased to be productive a long time ago.

In order not to overload the text with redundant data, three examples are given for each subordinator. Each example illustrates one of the three most common verbal forms in subordinates of immediate succession: past simple, present subjunctive, and present indicative.

The patterns that have been identified are described in the following subsections.

### 3.2.1. Preposition + que

Some subordinators such as *desde que* and *deque* are derived from the grammaticalization of an internal structure that follows the pattern P + *que* 'that', which comes from Vulgar Latin; see Herman (1963, p. 225) and Section 1.3. This pattern ceased to be productive to express immediate succession in the 16th century; see Eberenz (2014, § 34.9.1.3) and Herrero (2018). However, the subordinator *desde que* 'since', which, in present-day Spanish, introduces subordinates of initial delimitation, preserves the value of immediacy in Western Andalusian, Canarian, Dominican, and other Caribbean varieties, as in (20) (e.g., Eberenz 1982, pp. 339–40; 2014, p. 4251; Lope Blanch 1989, 1997; Herrero and González 1993; Herrero 2005, § 6.3.1.2; RAE-ASALE 2009, § 29.7r).

(20)  a.  [Desde que se enteró-IND.PASTP.3SG], le bloqueó en Facebook. (P.P., young woman, 17 May 2023, Santa Cruz de Tenerife, Canary Islands, Spain)
'As soon as she found out, she has blocked him on Facebook.'
   b.  [Desde que sepa-SUBJ.PRES.1SG que devolvieron todo lo robado], me comprometo a ir a la basilica (*sic*) a darle las gracias por la paz de esos ángeles. (*Twitter*: young man, 17 May 2023, Santo Domingo Norte, Dominican Republic)
'As soon as I know that everything stolen has been returned, I promise to go to the basilica to thank him for the peace of those angels.'
   c.  [Desde que entra-IND.PRES.3SG octubre], todo mejora (*Twitter*: young woman, 26 August 2023, Dominican Republic)
'As soon as October comes in, everything gets better.'

According to the data found in COSER (91 examples), *deque* 'as soon as', which has disappeared from standard Spanish, retains the meaning of immediacy that it had in Preclassical Spanish in rural varieties of Castile, Leon, Extremadura, Murcia, and the northwest of Andalusia, as in (21).

(21)  a.  [...] [de que llegó-IND.PASTP.3SG abajo a aquel prao], había una vaca con un ternerín [...]. ya... no lo mató, no, pero marchó por ahí abajo pa'l río y [de que llegó-IND.PASTP.3SG al río], encontró a una cerda con cerdines. (COSER-2644_01, old woman, 8 May 2004, Lucillo, León, Spain)
'As soon as he got down to that meadow, there was a cow with a calf. [...] he marched down there towards the river and as soon as he reached the river, he found a sow with piglets.'
   b.  [De que acabe-SUBJ.PRES.1SG de comer], me acostaré en siesta. (A.M., young woman, 2020, Ceclavín, Cáceres, Spain)
'As soon as I finish eating, I'll take a nap.'

It should be noted that the most common verbal forms combined with the subordinator *deque* are the indicative present (35 examples), in (22), and the past imperfect indicative with historical or narrative value (31 examples), in (23), because the speakers interviewed in COSER referred to activities that are repeated cyclically from year to year, as in (23).

(22)    a.    [De que se acababa-IND.PAST.IMP.3SG de segar aquí], pues hala, a segar pa ahí pa arriba, pa Sanabria, pa Galicia, pa ganar cinco duros o lo que fuera. (COSER-4622_01, old man, 9 May 2004, Mahide, Zamora, Spain) 'As soon as the mowing was finished here, then, to mow up there, to Sanabria, to Galicia, to earn five "duros" or whatever.'

          b.    [De que el guarro acababa-IND.PAST.IMP.3SG ya de sangrarse], pos cogían ya la sangre [. . .] y le echaban la cebolla y eran las primera(s) morcilla(s) que se llenaba(n). (COSER-7723_01, old woman, 17 April 2010, Orellana de la Sierra, Badajoz, Spain) 'When the pig had just bled to death, the blood was already dripping [. . .] and the onion was added, and they were the first blood sausage(s) to be filled.'

(23)    a.    Y [de que se enfría-IND.PRES.3SG], se corta, y si por dentro hay como. . . como unas huevitas en el jabón, está en lo punto. (COSER-1023_01, old woman, 21 May 2006, Campo Lugar, Cáceres, Spain) 'And as soon as it cools, it's cut, and if inside there are like. . . like little eggs in the soap, it's ready.'

          b.    Aquel pan gustaba, y este gusta el día primero, pero [deque pasan-IND.PRES.3PL dos días] no hay quien coma este pan de ahora. (COSER-2644_01, old woman, 8 May 2004, Lucillo, León, Spain) 'The bread of the past was liked, and the bread of the present is liked on the first day, but as soon as two days have passed, there is no one who can eat this bread.'

### 3.2.2. Adverb + que

Adv + *que* 'that' is a syntactic pattern inherited from Late Latin (see Herman 1963, p. 225; Hernández 2015; Herrero 2018) that continues to be productive today for coining new complex conjunctions of immediate succession. The subordinators analyzed in this section are *luego que*, *inmediatamente que*, and *enseguida que* 'as soon as', as well as some of their variants. In all of them, the adverb provides the immediacy value, while the conjunction *que* 'that' introduces the finite subordinate. Linguists such as Roy and Svenonius (2009), Den Dikken (2010), and Svenonius (2010), among others, suggest that the syntactic projection of the preposition *de* 'of' is always present, even if it does not appear explicitly in the syntax. This P is justified by the nominal properties of some adverbs, which allow them to assign the genitive case to their complements. The variant without the conjunction *que* is also documented heading participle (PART) and infinitive subordinates: *luego de* + PART / INF, *inmediatamente de* PART / + INF, and *enseguida de* PART + INF 'as soon as'.

The form *luego que* is the result of a grammaticalization process (*luego* + *que*) that expresses simple posteriority ('after') in most varieties of current Spanish, but it retains the immediacy value ('as soon as') that it had in Old Spanish when combined with the indicative past simple in cultured American texts, as in (24); see Herman (1963, p. 236), Eberenz (1982, pp. 358–60; 2014, p. 4226), Méndez (1995, p. 126), Barra (2002, pp. 298–305), Herrero (2005, pp. 250–53; 2018), and RAE-ASALE (2009, §31.14f).

(24)    a.    [. . .] [luego que se presentó-IND.PASTP.3SG el problema] se registró pérdida en la prisión. (CORPES XXI: Nelson Ardila Arias, www.eltiempo.com (accessed on 5 February 2023), 6 June 2001, Bogotá, Colombia) 'As soon as the problem occurred there was a loss of pressure.'

          b.    [Luego que se retiró-IND.PASTP.3SG el cliente], procedimos nuevamente a comunicarnos con el señor Pérez para investigar qué pasó. (CORPES XXI: Flor Díaz Davis, *Anécdotas bancarias*, 2005, Dominican Republic) 'As soon as the customer left, we contacted Mr. Perez again to find out what had happened.'

Its variant *luego de que* 'as soon as' has also been preserved in various American countries with the interpretation of immediate succession, as in (25).

(25) a. Hubo-IND.PASTP.3SG una crisis político y social [luego de que se denunció-IND.PASTP.3SG un presunto fraude electoral]. (CORPES XXI: Roberto Medina, www.larazon.com (accessed on 28 December 2022), 19 January 2023, Bolivia)
'There was a political and social crisis as soon as an alleged electoral fraud was denounced.'

b. La mucormicosis o el hongo negro, como también se le conoce, comenzó a generar preocupación en Venezuela [luego de que se confirmó-IND.PASTP.3SG el primer caso asociado al Covid 19 en el estado de Mérida]. (CORPES XXI: Luis de Jesús, www.elnacional.com (accessed on 5 February 2023), 28 August 2021, Caracas, Venezuela)
'Mucormycosis, or black fungus as it is also known, began to cause concern in Venezuela once it was confirmed the first case of Covid 19 in the state of Merida.'

The variant *luego de* + INF/+ PART, as in (26), can be documented in both American and European Spanish. However, the speakers of Peninsular Spanish with whom I have consulted these data tend to interpret the simple infinitive subordinates as cases of simple posteriority, while most agree on the meaning of the immediate succession of the perfect infinitive subordinates (*luego de haber dormido* 'immediately after having slept') and the participle subordinates (*luego de terminada la reunion* 'immediately after the meeting ended'). These speakers' interpretation of the data would indicate that, from a compositional point of view, it is the aspectual delimited character of the subordinate verb tense that is responsible for providing the immediacy component.

(26) a. [Luego de asentir-INF], el hombre habló: haré no importa qué por su señor padre. (CORPES XXI: Jorge Martínez Espinosa, *El final de los milagros*, 2001, Colombia)
'As soon as he nodded, the man spoke: I will do no matter what for his father.

b. Ambas expulsiones ocurrieron [luego de terminado-PART el partido]. (CORPES XXI: www.eltiempo.com (accessed on 5 April 2023), 4 July 2008, Bogotá, Colombia)
'Both ejections occurred after the end of the match.'

The subordinator *lueguito que* 'as soon as' appears in colloquial registers in Mexico, the Caribbean, Venezuela, Colombia, Peru, and other American countries. Although it can be easily documented on *Twitter*, as in (27a–b), no evidence is found, neither in CORPES XXI nor in PRESSEA. In the CdE WEBDIALETS *corpus*, there are four examples, but only two of them correspond to this subordinator. Finally, an example from the end of the 20th century can be found in CREA, in (27c).

(27) a. [Lueguito que salió-IND.PASP.3SG de la pileta], que se va acostando en una hamaca. (CREA: Eladia González, *Quién como Dios*, 1999, Mexico)
'As soon as he gets out of the pool, he lies down in a hammock.'

b. A su mamá la entambaron [lueguito que se alivió-IND.PASTP.3SG] y desde entonces la niña vive sola. (CdE WEBDIALECTS: http://dosdisparos.com (accessed on 6 September 2023), 8 April 2013, Chile)
'Her mother was imprisoned as soon as she gave birth and since then the girl has been living alone.'

c. [Lueguito que me acueste-SUBJ.PRES.1SG] empezaré a soñar. (A.C., young man, 25 May 2021, Ciudad de México, Mexico)
'As soon as I go to bed, I will begin to daydream.'

The American Spanish speakers with whom I was able to discuss examples of this subordinator do not agree in their interpretation. For some of them it is interpreted as the subordinator of simple posteriority, *después de que* 'after'. For others, however, the subordinator has a meaning that is equivalent to that of *apenas* 'as soon as'. In the first case, the diminutive has a purely affective value, while in the second, it is not clear whether the value of immediate succession comes from the adjunction of the diminutive or whether it is inherited from the adverb *luego* 'after'. A quantitative study with a larger number of speakers should be carried out to find out which interpretation corresponds to each example to draw reliable conclusions.

The subordinator *luego luego que* 'as soon as' (Mexico) is derived from the lexical reduplication of the adverb *luego* 'after'. Lexical reduplication is a grammatical phenomenon by which a lexical word X is repeated, obtaining a segment XX whose meaning is different from that of the single unit X. The interpretations that lexical reduplication can give rise to are different depending on the lexical category that is duplicated: intensification, narrowing, expansion, or affective meanings; see Moravcsik (1978), Escandell (1991), García Page (1997), Roca and Suñer (1997–1998), Ghomeshi et al. (2004) and Feliu (2011), among others. The reduplication of adverbs, in (28), results in a superlative meaning in Spanish.

(28)  a.  Habla rápido.
         '(S)he speaks quickly.'
     b.  Habla rápido rápido.
         '(S)he speaks very quickly.'

The construction of the meaning of immediate succession in the internal structure of the subordinator *luego luego que* 'as soon as' is transparent since it is obtained from the lexical reduplication of the adverb *luego* 'after', which indicates simple posteriority. Subsequently, the reduplicated adverb *luego luego*, which exists independently in the language with the meaning 'immediately', is merged with the general subordinator *que* 'that', in (24); see Academia Mexicana de la lengua (2010).

(29)  a.  [Luego luego que me desperté-IND.PASTP.1SG], cerré los ojos para que no se me escapara y me quedé un buen rato aprendiéndome el sueño de memoria. (CORPES XXI: Alejandro Carrillo Rosas, *Adiós a Dylan*, 2016, Mexico)
         'As soon as I woke up, I closed my eyes so that I wouldn't miss it, and I stayed for a long time, memorizing the dream.'
     b.  Mi esposa es de las que si no se duerme [luego luego que se acuesta-IND.PRES.3SG] prende la tele. (*Twitter*: young man, 8 December 2020, Ciudad de México, Mexico)
         'My wife is the type that if she doesn't go to sleep as soon as she goes to bed, she turns on the TV.'
     c.  Pinches simios los que se paran [luego luego que aterriza-IND.PRES.3SG el avión]. (*Twitter*: young man, 9 August 2023, Ciudad de México, Mexico)
         'Fucking apes who stand up as soon as the plane lands.'

*Luego luego que* 'as soon as', in (29), only appears five times in CORPES XXI and five times in CdE WEBDIALECTS, although it is easily documented in Mexican *twits*.

The subordinator *enantito que* 'as soon as' (colloquial language of Panama) comes from the Old Spanish adverb *enante(s)* 'recently', which is still used in some American countries, especially in Panama, Colombia, and Venezuela, as well as in the Caribbean area and in some parts of the Andean region; see RAE-ASALE (2009, § 30.6q) and Pato (2019). The meaning of immediacy of *enantito que* 'as soon as', in (30), is obtained through the amalgamation of the generic subordinator *que* 'that' with the adverb of immediacy *enantito*.

(30)  a.  Y [enantito que la vi-IND.PASTP.1SG] la busque (*sic*) en *fb* y me borro (*sic*) de su vida: (*Twitter*: young man, 31 March 2014, Alcalde Díaz, Panama)
         'As soon as I saw her, I looked her up on facebook and she deleted me from her life.'
     b.  [Enantito que salgo-IND.PRES.1SG a comprar comida de una vez] cae el aguaceron (*sic*). (*Twitter*: young man, 10 November 2015, Ciudad de Panamá, Panama)
         'As soon as I go out to buy food the big downpour comes down.'

The data for *enantito que* were extracted entirely from *Twitter* since this form does not appear in the *corpora* CORPES XXI, PRESSEA, and CdE WEBDIALECTS.

According to Eberenz (1982, pp. 357–78), *inmediatamente que* 'as soon as' appeared in the middle of the 19th century. However, examples in CORDES and CORDIAM *corpora* allow us to date its origin to the middle of the 18th century; see Herrero (2017). Although it is possible to find examples in European Spanish (eight examples in CORPES XXI), this form is preferred in America (39 examples in CORPES XXI). The vitality of *inmediatamente que* 'as

soon as' on *Twitter* confirms its consolidation in all registers of contemporary colloquial American Spanish, as in (31).

(31) a. [Inmediatamente que terminó-IND.PASTP.3SG el curso] se aplicó nuevamente al cuestionario. (CORPES XXI: Iris Cortina Mena and Bertha Bello Rodríguez, www.revmatanzas.sld.cu (accessed on 1 June 2023), 1 April 2008, Cuba)
'As soon as the course ended the questionnaire was applied again.'

  b. Ponte buza con lo de los boletos para que los compremos [inmediatamente que salgan-SUBJ.PRES.3PL] (*Twitter*: young woman, 22 February 2022, Ciudad de México, Mexico)
'Keep an eye out for tickets so we can buy them as soon as they come out.'

  c. Es de pinches obesos levantarse [inmediatamente que entran-IND.PL.3PL los snacks en una sala de juntas]. (*Twitter*: young man, 7 November 2022, Ciudad de México, Mexico)
'It's like a fucking obese person to get up as soon as snacks enter a boardroom.'

Like other subordinators built from the Adverb + *que* scheme, *inmediatamente que* 'as soon as' has the *inmediatamente de que* variant with the same meaning, which is also used in American Spanish (two examples in CORPES XXI), as in (32).

(32) a. Es digno de mencionar que [inmediatamente de que ocurrió-IND.PASTP.3SG el siniestro], el Jefe del Gobierno de D.F. [...] llegó a las instalaciones de esa unidad para colaborar con las autoridades federales. (CORPES XXI: www.ideas4solutions.net (accessed on 15 May 2023), 4 February 2013, Colombia)
'It is worth mentioning that as soon as the incident occurred the head of the government of Mexico City [...] arrived at the facilities of this unit to cooperate with the federal authorities.'

  b. [Inmediatamente de que su integridad ya no esté-SUBJ.PRES.3SG en peligro], debe esposar o inmovilizar al civil. (*Twitter*: young man, 29 January 2023, Tlaquepaque, Jalisco, Mexico)
'As soon as your integrity is no longer in danger], you must handcuff or immobilize the civilian.'

  c. [Inmediatamente de que sale-IND.PRES.3SG la oferta], me pongo a hacer chingos de horarios para terminar improvisando uno en la última media hora. (*Twitter*: young man, 10 November 2022, Xalapa, Veracruz de Ignacio de la Llave, Mexico)
'As soon as the offer comes out, I start making a bunch of schedules and end up improvising one in the last half hour.'

The *inmediatamente de* 'as soon as' variant is documented in nine examples in CORPES XXI before an infinitive subordinate clause and in seven examples before a past participle subordinate clause, as in (33).

(33) a. [Inmediatamente de colocar-INF la prueba nasofaríngea en el test], la persona estará en capacidad de saber si está o no contagiada con el Covid. (CORPES XXI: Basyl Macías, www.el-carabobeno.com (accessed on 17 May 2023), 24 March 2020, Venezuela)
'Immediately after placing the nasopharyngeal swab on the test, the person will be able to know whether or not he/she is infected with Covid.'

  b. [Inmediatamente de terminado-PART el ensayo], el señor Brugo puso en manos del señor Ministro del Interior una solicitud del ingeniero Strante. (CORPES XXI: Aníbal Orué Pozzo, *Comunicación y estado*, 2003, Paraguay)
'Immediately after the trial ended, Mr. Brugo handed to the Minister of the Interior a request from engineer Strante.'

*Enseguida que* 'as soon as', in (34), appears in the mid-18th century and may be documented in all varieties of Spanish (e.g., RAE-ASALE 2009, 31.14e and 31.14i, and Herrero 2017, 2018), but it does not reach a high frequency of use in any of them (there are eight examples from European Spanish and twenty-one examples from American Spanish in CORPES XXI). Its variant *enseguida de que* 'as soon as', in (35), is characteristic of

American Spanish (nineteen examples in American Spanish/zero examples in European Spanish in CORPES XXI).

(34)   a.   [Enseguida que llegué-IND.PASTP.1SG] pedí hablar con la directora. (CORPES XXI: Benigno Dou, *Luna rota*, 2002, Venezuela)
'As soon as I arrived, I asked to speak to the director.'

b.   Es tan lindo este *anime*, de aquí me paso al manga [enseguida que termine-SUBJ.PRES.1SG el *anime* jj] (*Twitter*: young woman, 28 July 2023, Cuba)
'It's so cute this anime, I'm moving on to the manga from here as soon as I finish the anime jj'

c.   Es tan difícil no abrir tus mensajes [enseguida que me llegan-IND.PRES.3PL]. (*Twitter*: young woman, 26 July 2023, Barranquilla, Colombia)
'It's so hard not to open your messages as soon as they reach me.'

(35)   a.   No esperaba que Natalia lo usara [enseguida de que Pablo la protegió-IND.PASTP.3SG]. (*Twitter*: young woman, 27 June 2023, Ciudad de México, Mexico)
'I didn't expect Natalia to use it once Pablo protected her.'

b.   [Enseguida de que acabe-SUBJ.PRES.1SG], le riego las plantas. (J.C., young man, 16 May 2023, Cuernavaca, Morelos, Mexico)
'As soon as I finish, I water the plants.'

c.   Tampoco se tiene que arrugar si la tiendes [enseguida de que sale-IND.PRES.3SG de la lavadora]. (*Twitter*: young woman, 19 June 2023, Ciudad de México, Mexico)
'Nor should it wrinkle when hung once it comes out of the washing machine.'

The variant *enseguida de* 'as soon as' can head infinitive subordinate clauses, as in (36a–b), or participle subordinate clauses, as in (36c). In CORPES XXI, there are twenty-two examples of infinitive subordinate clauses (three in European Spanish and nineteen in American Spanish), and there is only one example with a participle subordinate clause.

(36)   a.   Regresaron al campamento en silencio, contentos, y se separaron [enseguida de cruzar-INF el puente]. (CORPES XXI: *Miguel Morra, Los días del agua*, 2003, Uruguay)
'They walked back to the camp in silence, satisfied, and parted as soon as they had crossed the bridge.'

b.   Lo supe [enseguida de conocerte-INF], al ver cómo tratabas a mis amigos y a las novias de tus compañeros. (CORPES XXI: Berta Marsé, *En jaque*, 2006, Spain)
'I knew it when I met you, seeing how you treated my friends and your classmates' girlfriends.'

c.   [Casi enseguida de llegados-PART a París] fuimos a visitar a mi tía Manolita, la esposa de mi tío Segundo. (CORPES XXI: Carlos Blanco Aguinaga, *De mal asiento*, 2010, España)
'Almost as soon as we arrived in Paris, we went to visit my Aunt Manolita, my Uncle Segundo's wife.'

Some adverbial subordinators of immediate succession, such as *apenas*, *no bien, and ni bien* 'as soon as', which are described in Section 3.2.3, alternate in very colloquial registers of American Spanish with the variants *apenas que*, *no bien que*, and *ni bien que* 'as soon as', which have the same meaning, as in (37). Neither in CORPES XXI nor in PRESSEA nor in the CdE WEBDIALECTS have I found any examples of these uses, but these variants are easily documented on *Twitter*.

(37)   a.    Mi perro viéndose todo malote, pero [apenas que llega-IND.PRES.3SG un
desconocido] comienza a mover la colita y a tirarse a sus pies. (*Twitter*: young
man, 1 May 2021, Ciudad de México, Mexico)
'My dog looks very tough, but as soon as a stranger arrives, he starts wagging his
tail and jumping at his feet.'

   b.    [No bien que pongo-IND.PRES.1SG la bolsa en su lugar], mi gatita levanta la
cabeza. (*Twitter*: young woman, 4 February 2020, González Catán, Argentina)
'As soon as I put the bag in place, my kitten lifts her head.'

   c.    [Ni bien que digan-SUBJ.PRES.3PL que es falso], lo borro directamente. (*Twitter*:
young woman, 19 November 2023, Buenos Aires, Argentina)
'As soon as they say it's fake, I delete it immediately.'

### 3.2.3. Adverb + VFINITE

This section contains a series of heterogeneous elements (negative polarity terms, negations, approximatives, aspectual or focus adverbs, etc.) which have in common the fact that they are derived from modifiers and adjuncts of the subordinate clause that, through a process of grammaticalization, become subordinators heading participle/finite subordinates of immediate succession; see Brucart and Gallego ([2009] 2016) and Haegeman (2012, chap. 5).

Although they come from different categories, the subordinators *apenas*, in (38), *no bien*, in (39), and the variant *ni bien* 'as soon as' (in Argentina, Uruguay, and Peru), in (40), have in common the fact that they highlight the imminent culmination of the first event.

(38)   a.    Lo compré [apenas salió-IND.PASTP.3SG al mercado] y lo guardé hasta hoy.
(*Twitter*: old man, 30 July 2023, Buenos Aires, Argentina)
'I bought it as soon as it went on sale and kept it until today.'

   b.    Unos quince mil. Présteme y le prometo que [apenas salga-SUBJ.PRES.1SG de aquí]
se los pago. (CORPES XXI: Ricardo Raphael, *Hijo de la guerra*, 2019, México)
'About fifteen thousand. Lend me and I promise that as soon as I leave here, I'll
pay you back.'

   c.    ¿Qué hacen [apenas se despiertan-IND.PRES.3PL]? (*Twitter*: young man, 23
December 2021, Colombia)
'What do they do as soon as they wake up?'

(39)   a.    [No bien arrancó-IND.PASTP.3SG el ómnibus], ellos se sentaron y yo me distraje
mirando por la ventanilla. (CORPES XXI: Carlos Dámaso Martínez, *El amor cambia*,
2001, Argentina)
'As soon as the bus started, they sat down, and I was distracted looking out of
the window.'

   b.    [No bien termine-SUBJ.PRES.3SG el avituallamiento], zarpa la flota. (CORPES XXI:
Adelaida Fernández Ochoa, *Afuera crece un mundo*, 2017, Colombia)
'As soon as the victualling is finished, the fleet sails.'

   c.    [No bien se reconocen-IND.PRES.3PL], los dos jóvenes se saludan con afecto.
(CORPES XXI: Antonio Orejudo, *La casa de los Peláez. La historia completa*,
2020, Spain)
'As soon as they recognize each other, the two young men greet each
other warmly.'

(40)   a.    [Ni bien salió-IND.PASTP.3SG], preguntó a uno de los guardias dónde quedaba la
agencia de Varig. (CORPES XXI: Roberto Paredes, *El Somozano: Novela sobre un
ajusticiamiento*, 2005, Paraguay)
'As soon as he left, he asked one of the guards where Varig's agency was located.'

   b.    [Ni bien logres-SUBJ.PRES.2SG recuperar el acceso a tu cuenta], activá el segundo
factor de autenticación. (CORPES XXI: www.lanacion.com.py (accessed on 5
September 2022), 17 June 2021, Paraguay)
'As soon as you regain access to your account, activate the second
authentication factor.'

   c.    [Ni bien llegás-IND.PASTP.3SG], la Richard se pone chocha. (A.D., middle-aged
woman, 11 November 2008, Buenos Aires, Argentina)
'As soon as you get there, la Richards gets horny.'

According to the data found in CORDE, the first occurrences of *apenas* 'as soon as', and much less frequently *no bien* 'as soon as', with subordinative value appeared in the Spanish Golden Age ("Siglo de Oro") in adverbial past participle subordinates, which persist in the current language, as in (41); see Suñer (2014).

(41)   a.   [Apenas terminada-PART la instrucción primaria] comenzó a trabajar en el taller mecánico de Steve Toil. (CORPES XXI: Carlos Rubio Rosell, *Los Ángeles-Sur*, 2001, México)
'As soon as he finished elementary school, he started working in Steve Toil's machine shop.'

       b.   [No bien comenzado-PART el *show* con el tango de Mores y Manzi *Una lágrima tuya*], desde el escenario salió el humo típico de todo espectáculo. (CORPES XXI: Karina Micheletto, www.pagina12.com.ar (accessed on 23 February 2023), 27 February 2005, Argentina).
'As soon as the show began with the tango of Mores and Manzi *Una lágrima tuya*, the typical smoke of each show rose from the stage.'

*Apenas* and *no bien* 'as soon as' coexist in current Spanish with homophone forms that are in intermediate phases of a grammaticalization process so that the grammaticalization paths of these subordinators can be easily reconstructed. The process begins when these elements appear in correlation with an inverse or narrative *cuando* 'when', as in (42) (e.g., Eberenz 1982, pp. 317–19; 2014, § 34.1.5.4; García Fernández 2000, p. 248; Pavón 2013; RAE-ASALE 2009, § 23.12o, § 24.50-t).

(42)   a.   *Apenas* colgó-IND.PASTP.3SG *cuando* Falfaro abría la suntuosa puerta del despacho presidencial y dejaba-IND.PAST.IMP.3SG entrar a una nube de periodistas.
(CORPES XXI: Jorge Maronna y Luis María Pescetti, *Copyright: Plagios literarios y poder político al desnudo*, 2001, Argentina)
'He had barely hung up when Falfaro opened the ornate door of the president's office and let in a cloud of journalists.'

       b.   *No bien* acababa-IND.PAST.IMP.1SG de *decirlo* cuando Rodrigo empezó a vociferar.
(CORPES XXI: Rafael Tovar y de Teresa, *Paraíso es tu memoria*, 2012, México)
'No sooner had she finished saying it when Rodrigo start shouting.'

The consolidation of *apenas* and *no bien* 'as soon as' as subordinators occurred when the inverse *cuando* was omitted (late 18th century), and they can head a subordinate by themselves.

In the diachronic *corpora* CORDE and CORDIAM, there are no examples of the correlation of *ni bien* with inverse *cuando*, which is very common for both *apenas* and *no bien*. Although this question should be studied in more detail than can be offered here, the subordinator *ni bien*—first documented in CORDE from Argentina, 1872—seems to have originated via analogy with *no bien* in countries of the South American cone.

In current Spanish, the forms *apenas, no bien,* and *ni bien* 'as soon as' are in the process of expanding their contexts since they can head infinitive subordinates, as in (43), probably via analogy with other immediate subordinates such as *nada más* y *nomás* 'as soon as' (see Section 3.2.4).

(43)   a.   ¡No puede ser legal que haga tanto calor, [apenas empezar-INF el día], viejo!
(*Twitter*: young man, 22 December 2021, Garauhape, Misiones, Argentina)
'It can't be legal for it to be so hot at the start of the day, old man.'

       b.   [No bien entrar-INF a la preadolescencia], mis chilpayates eligen afinidades contrarias a las mías. (*Twitter*: young woman, 15 September 2021, Santiago de Querétaro, Mexico)
'As soon as I enter pre-adolescence, my children choose affinities contrary to mine.'

       c.   [. . .] casi me levanto y me voy [ni bien empezar-INF]. (*Twitter*: young man, 31 July 2023, Buenos Aires, Argentina)
'I almost got up and left as soon as I started.'

In European Spanish, *apenas* 'as soon as' is mostly documented in written texts and usually does not appear in the spoken language, where *en cuanto* 'as soon as' is the preferred option. In contrast, in American Spanish, especially in Peru and Colombia, apenas 'as soon as' is used in both written and spoken language; see Herrero (2018),

In the colloquial registers of *Twitter*, mainly in American Spanish, *apenas*, *no bien*, and *ni bien* are also used in correlation with the conjunction *y*, as in (44).

(44)  a.  *Apenas* llegó-IND.PASTP.3SG *y* ya le dio alegría a todo el estudio. (*Twitter*: young woman, 16 November 2023, Buenos Aires, Argentina)
S/He's just arrived, and s/he's already brought joy to the entire studio.'
  b.  Uno *no bien* llega-IND-PRES-3SING *y* ya están preguntando cuándo uno se va. (*Twitter*: young woman, 26 July 2022, Santo Domingo, Dominican Republic)
'You have just arrived and you are already being asked when you are going to leave.'
  c.  Ni bien llega-IND.PRES.3SG El Niño *y* ya se especula con el precio del arroz. (*Twitter*: young man, 12 June 2023, Guayaquil, Ecuador)
'El Niño has barely arrived, and rice prices are already the subject of speculation.'

In these cases, which should be studied in more detail, *apenas*, *no bien* and *ni bien* would be modifiers of the first sentence since they have not completed their grammaticalization processes as subordinators; see Pavón and Suñer (2022).

The subordinator *recién* 'as soon as' is currently undergoing a process of grammaticalization. In CORPES XXI, there is only one example combined with a finite verb (VFINITE), shown in (41a), although it can be easily documented on *Twitter* in Rioplatense, Chilean, Andean, Uruguayan, Colombian, and Peruvian Spanish, as in (45b–c).

(45)  a.  [Recién entré-IND.PASTP.1SG] me tocó matar a una persona. (CORPES XXI: Guillermo González Uribe, *Los niños de la Guerra*, 2002, Bogotá, Colombia)
'As soon as came in, I had to kill a person.'
  b.  [Recién salí-IND.PASTP.1SG de casa], empezó a llover. (F.G. young man, 11 November 2022, Mendoza, Argentina)
'As soon as I left the house, it started to rain.'
  c.  [Recién amaneció-IND-PASTP.3SG], la mandamos de una patada a audicionar para la empresa más bonita de todas. (*Twitter*: young woman, 19 March 2021, Lima, Peru)
'As soon as the dawn broke, we sent her off to audition for the most beautiful company of all.'

The subordinator comes from the adverb *recién* 'shortly before' (see Ramírez Luengo 2007), which, in many varieties of American Spanish, may be a modifier of a finite V to indicate that the event has just happened. The first cases of *recién*-MODIFIER to *recién*-SUBORDINATOR reanalysis occur in adverbial participle subordinates, which can be documented for the first time in Spanish at the end of the 17th century and beginning of the 18th century (1710: first documentation in CORDE, Argentina; 1682: first documentation in CORDIAM, Mexico); see Suñer (2014). These constructions have remained in current American Spanish, as in (46).

(46)  a.  [Recién llegada-PART a París], mi esposa contrajo una tromboflebitis y el médico dijo que debía permanecer internada por unas cuantas horas. (CORPES XXI: Alfredo Bryce Echenique, *Permiso para retirarme, Antimemorias III*, 2021, Peru)
'Having just arrived in Paris, my wife came down with thrombophlebitis and the doctor said she needed to stay in the hospital for a few hours.'
  b.  [Recién asentado-PART en su nuevo espacio radial], nos cuenta sobre los saltos de la fe, la soledad, la deconstrucción del macho y el amor en todas sus formas. (CORPES XXI: Soledad Simond, www.lanacion.com.ar (accessed on 28 February 2023), 30 February 2021, Argentina)
'Recently settled in his new radio space, he tells us about leaps of faith, loneliness, the deconstruction of the masculine, and love in all its forms.'

From a compositional point of view, the value of the immediate succession of *recién* + PART comes from the combination of the adverb, which emphasizes the culmination of the first event, and the use of the past participle morphology, which refers to a completed event.

*Recién* 'as soon as' has not consolidated its grammaticalization process as a subordinator of adverbial finite sentences since it is only combined with the past simple of the indicative, as in (45), cf. § 1.2. On the other hand, it is easy to document examples in which *recién* precedes the generic temporal subordinator *cuando* (225 examples in CORPES XXI), as in (47a), or follows it (289 examples in CORPES XXI), as in (47b), and is used in constructions with inverse *cuando*, as in (47c).

(47)    a.    [Recién cuando bajé-IND.PASTP.1SG del auto] me di cuenta de que la puerta de lado estaba chocada. (CORPES XXI: Ana María Shua, *Historias verdaderas*, 2004, Argentina)
'Only when I got out of the car did I realize the side door was smashed.

        b.    Estabas muy flaca [cuando recién viniste-IND.PASTP.2SG de Posadas]. (*Twitter*: young woman, 7 January 2019, Puerto Iguazú-Posadas, Argentina)
'You were very skinny when you first came from Posadas.'

        c.    *Recién* me puse-IND.PRES.1SG a llorar *cuando* su hermano más chico, que era compañero de mi hermana Debi, volvió al colegio de la mano de su mamá. (CORPES XXI: Tamara Tenenbaum, *Todas nuestras maldiciones se cumplieron*, 2021, Argentina)
'I just started crying when his younger brother, who was my sister Debi's classmate, came back to school hand in hand with his mom.'

Herrero (2018) points out that the most probable intermediate structures to obtain the subordinator *recién* 'as soon as', the construction of inverse *cuando*, as in (47c), and those constructions in which *cuando* 'when' precedes the aspectual modifier *recién* 'just' 'recently', as in (47b), since the geographical area where the combination *recien cuando*, as in (47a), is documented do not coincide with the places where the first cases of the subordinator occur. Herrero also notes that in the constructions of *recién cuando*, as in (47a), *recién* has a focal value with a meaning equivalent to *solo* 'only', which is reflected in the gloss of (47a).

3.2.4. Adverb + INF

According to the data found in CORDE and CREA, the construction *nada más (de)* + INF 'as soon as' appeared in European Spanish in the mid-20th century. It may have also been documented simultaneously in some American varieties, especially in Mexican and Caribbean Spanish. This subordinator still exists today in both European and American Spanish, as in (48).

(48)    a.    Tendría que haberlo hecho [nada más salir-INF de Ibiza]. (CORPES XXI: Kiko Amat, *Revancha*, 2021, Spain)
'I should have done it [as soon as I left the Ibiza].

        b.    [Nada más de verte-INF], sonrío-IND.PRES.1SG. (A.C., young man, 17 May 2019, Ciudad de México, Mexico)
'As soon as I see you, I have a smile on my face.'

In the CREA *corpus*, the subordinator *nomás* (*de*) + inf 'as soon as' appeared for the first time in 1975. Although it presumably existed earlier in the spoken language, I could find no example in either CORDE or CORDIAM. According to data from CORPES XXI (22 examples) and *Twitter*, *nomás* (*de*) + INF is fully consolidated in current Mexican Spanish, as in (49).

(49)    a.    [Nomás acabar-INF la gala], me llamó por teléfono para decirme que me quería mucho. (*Twitter*: young man, 30 January 2023, Ciudad de México, Mexico)
'As soon as the gala was over, he called me to tell me that he loved me very much.'

        b.    [Nomás de entrar-INF a Twitter] me llega el olor a millennials frágiles (*Twitter*: young man, 17 May 2019, Ciudad de México, Mexico)
'As soon as I log on to Twitter, I get the smell of the fragile millennials.'

The INF constructions introduced by *nada más* and *nomás* 'as soon as' alternate with subordinates headed by complex conjunctions such as *nada más que*, as in (50), and *nomás que* (51).

(50)   a.   La compraré [nada más que salga-SUBJ.PRES.1SG]. (*Twitter*: young man, 21 July 2023, Oviedo, Spain)
           'I will buy it [as soon as it comes out].'
       b.   [Nada más que te suelten-SUBJ.PRES.3PL], te vienes para aquí conmigo. (CORPES XXI: Álvaro Pombo, *Santander-1936*, 2023, Spain)
           'As soon as they let you go, come with me.'

(51)   a.   [Nomás que acabe-SUBJ.PRES.1SG mi carrera], me voy a comprar un rancho chingón. (*Twitter*: young woman, 13 September 2019, Ciudad de México, Mexico)
           'As soon as I finish my degree, I'm going to buy a cool ranch.'
       b.   [Nomás que mi amá me encuentre-SUBJ.PRES.3SG en tik tok], van a llover madrazos! (*Twitter*: young woman, 15 November 2023, Puebla, México)
           'As soon as my mom finds me in Tik Tok, it's going to rain down blows.'

*Nada más que* 'as soon as' appears four times in CORPES XXI (two in Spain and two in Mexico), but it is easy to document it on *Twitter*. It is currently under consolidation since it is mostly constructed with the present subjunctive. I have not found cases where it is combined with the present indicative or in other verb tenses. CORPES XXI does not include any cases of *nomás que* 'as soon as'. Although some examples can be found on *Twitter* (from Mexico and other American countries), they are much less numerous than those of the complex conjunction *nada más que*, 'as soon as'. Even rarer are the examples where *nada más que* 'as soon as' and *nomás* 'as soon as' head finite subordinates of immediate succession without the complementizer *que* 'that', as in (52) and (53), which I could only find on *Twitter*.

(52)   a.   [Nada más se vaya-SUBJ.PRES.3SG Maxi], vendrá Arthur. (*Twitter*: young man, 16 August 2022, Elche, Spain)
           'As soon as Maxi leaves, Arthur will come.'
       b.   Tranquilo, que [nada más salga-SUBJ.PRES.3SG] solo estará al doble de precio en Wallapop. (*Twitter*: young man, 9 August 2023, Valencia, Spain)
           'Don't worry, as soon as it comes ou] it will only be twice the price on Wallapop.'

(53)   a.   [Nomás acabe-SUBJ.PRES.3SG la cuarentena], me pongo peda para poder besarte. (*Twitter*: young woman, 5 October 2020, Puebla, Mexico)
           'As soon as the quarantine is over, I'll get drunk so I can kiss you.'
       b.   [Nomás se duerma-SUBJ.PRES.3SG el chamaco], escapo y veo la tele un ratito. (R.R., young woman, 28 July 2023, Mérida, Oaxaca, México)
           'As soon as the kid falls asleep, I run away and watch TV for a while.'

On *Twitter*, the hybrid construction *al nomás* + INF 'as soon as' (only one example in CORPES XXI) is geographically located in Mexico, Guatemala, Honduras, El Salvador, Colombia, Venezuela, and other American countries, as in (54).

(54)   a.   ¿Adivinen quién se dormirá [al nomás sentarse-INF en el avión]? (*Twitter*: young man, 26 July 2019, Ciudad de Guatemala, Guatemala)
           'Guess who falls asleep as soon as he sits down on the plane?'
       b.   [Al nomás despertar-INF], tómate un vaso, otro antes de cada comida. (*Twitter*: young woman, 15 November 2023, Santa Ana, El Salvador)
           'As soon as you wake up, drink a glass, another one before each meal.'

The subordinator *al nomás que* 'as soon as' does not appear in *corpora* although, according to data extracted from *Twitter*, it is quite common in colloquial speech in Mexico, Central America, Argentina, and other American countries. The data show that it is consolidated with the three most common verb forms in which subordinates of immediate succession appear, as in (55).

(55)  a.  ¿Por qué lo corrieron [al nomás que lo vieron-IND.PASTP.3PL]? (*Twitter*: young man, 28 January 2023, El Salvador)
'Why did they throw him out as soon as they saw him?'

b.  Tengan paciencia de un solo va a arreglar todo el centro [al nomás que acaben-SUBJ.PRES.3PL de arreglar toda (sic) las tuberías que están arreglando]. (*Twitter*: young man, 19 August 2023, El Salvador)
'Be patient, in a short time, the whole center will be fixed as soon as they finish fixing all the pipes they are fixing.'

c.  [...] dice que se duerme [al nomás que su cabeza toca-IND.PRES.3SG su almohada]. (young man, 30 January 2023, México)
'(S)he says he falls asleep as soon as his head touches his/her pillow.'

The meaning of the immediate succession of *al nomás* + inf 'as soon as', and its variant *al nomás que* 'as soon as', is obtained compositionally from the syntactic pattern *al* + infinitive, which expresses simple posteriority, together with the aspectual/focal modifier *nomás*, which provides the component of immediacy.

3.2.5. Modal and Quantitative Structures

This section includes the analysis of different subordinators (*tan pronto como* > *tan pronto* 'as soon as; *tan luego como* > *tan luego* 'as soon as') that have in common the fact of including, in their internal structure, a quantifier that expresses the measure of the short interval that separates two successive events. The subordinator *en cuanto* 'as soon as' and its variants *cuanto que* 'as soon as' and *en cuanto que* 'as soon as', described in Section 3.1, should also be included in this section because the quantifier *cuanto* expresses the measure of the interval. Finally, the subordinator *así que* 'as soon as', which contains a deictic element signaling the end point of the first event, is also examined in this section.

In Old Spanish, as well as in current Spanish, the internal structures of some subordinators resulted from the grammaticalization of a comparative correlation of equality, as in (56).

(56)  [$_{SQ}$ [$_{Q\ COMPARATIVE}$ *tan*] + [$_{ADV}$ Comparative Dimension] + [$_{CODA}$ as [Subordinate]]].

The comparative dimension is embodied in an adverb indicating proximity in time. In Medieval Castilian, the most common correlative was *tan aína como* 'as soon as', while from Classical Spanish until the 18th century, the dominant form was *tan presto como* 'as soon as', cf. Herrero (2018). The subordinator of present-day Spanish, *tan pronto como* 'as soon as' (57), is the result of the grammaticalization of the comparative correlation. It emerged in the 17th century via analogy with earlier structures and has been documented in both European and American Spanish.

(57)  a.  [Tan pronto como se terminó-IND.PASTP.3SG la copa], volvió a subir a escondidas. (CORPES XXI: Claudia Pradas Gallardo, *Todo saldrá (bien)*, 2022, Spain)
'As soon as the drink was finished, he sneaked back upstairs.'

b.  En Turquía se considera de buena suerte rociar sal en la puerta de las casas [tan pronto como el reloj marca-IND.PRES.3SG la medianoche del día de Año Nuevo]. (CORPES XXI: www.lapatilla.com (accessed on 15 June 2023), 31 December 2022, Venezuela)
'In Turkey it is considered good luck to sprinkle salt on the door of houses [as soon as the clock strikes midnight on New Year's Day.'

c.  Aunque muchas de las dietas pueden ayudar a perder peso mientras se siguen, [tan pronto como se reanude-SUBJ.PRES.3SG el sistema de vida habitual, si este no es el adecuado], la subida de peso está de nuevo garantizada. (CORPES XXI: Mercè Palau, www.eldiario.es (accessed on 15 June 2023), 10 August 2021, Spain)
'Although many of the diets may help to lose weight while they are being followed, as soon as the usual lifestyle is resumed, if it is not adequate, weight gain is again guaranteed.'

The variant *tan pronto* 'as soon as', as in (58), which has the same meaning as *tan pronto como* 'as soon as', appeared at the beginning of the 20th century. The loss of *como* 'as' is clear evidence that the grammaticalization process of the internal structure of the subordinator has been completed since the speaker is unaware of the comparative value of the initial structure.

(58)　　a.　　[Tan pronto me acostumbré-IND.PASTP.1SG a la luz roja], se volvió blanca de nuevo. (CORPES XXI: Rodrigo Cortés, *Los años extraordinarios*, 2021, Spain)
'As soon as I got used to the red light, it turned white again.'

　　　　b.　　[Tan pronto tengamos-SUBJ.PRES.1PL una información oficial], la vamos a difundir. (CORPES XXI: Sandra Guzmán, www.diariolibre.com (accessed on 10 June 2023), 5 August 2021, Santo Domingo, Dominican Republic)
'As soon as we have official information, we will disseminate it.'

　　　　c.　　[Tan pronto llega-IND.PRES.3SG de sus clases], toma un refrigerio, se da un baño y comienza de inmediato a revisar lo que estudió en el día. (CORPES XXI: Ángel Cintrón Opio, "El dilema de las asignaciones", www.adendi.com (accessed on 10 June 2023), 3 February 2009, Puerto Rico)
'As soon as Ana comes home from class, has a snack, takes a bath, and immediately begins to review what she learned during the day.'

*Tan luego como* 'as soon as', in (59), was created through analogy with *tan pronto como* 'as soon as' in the mid-18th century. Herrero (2018, p. 773) points out that this subordinator lost its vitality during the 20th century since it has not been documented with the same frequency as in the previous century in CORDE, CREA, and PRESSEA and only three examples appear in CORPES XXI. However, the abundance of examples of this subordinator on *Twitter* suggests that this form has remained in the colloquial language of Mexico.

(59)　　a.　　[Tan luego como la izquierda llegó-IND.PASTP.3SG al Congreso] se podía escuchar otra manera de ser, pensar, investigar, analizar y proponer. (CORPES XXI: Pablo Gómez, www.proceso.com.mx (accessed on 12 June 2023), 13 March 2020, Mexico)
'As soon as the left came to Congress, another way of being, thinking, investigating, analyzing and proposing could be heard.'

　　　　b.　　Difícil para Amlo (Andrés Manuel López Obrador) encontrarle un jale de alto nivel a Yasmín, [tan luego como pierda-SUBJ.PRES.3SG el cargo en la SCJN (Suprema Corte de Justicia de la Nación)] (*Twitter*: young man, 2 March 2023, Ciudad de México, Mexico)
'It will be difficult for Amlo (Andrés Manuel López Obrador) to find a high-level work for Yasmín, as soon as she loses her position at the SCJN (Supreme Court of Justice of the Nation).'

　　　　c.　　Yo no les sigo el juego, [tan luego como leo-IND.PRES.1SG su *twit*], no contesto, bloqueo. (*Twitter*: young man, 11 January 2022, Ciudad de México, México)
'I don't play along, as soon as I read their twit, I don't answer, I block.'

Herrero (2018, p. 773) indicates that the form *así que* 'as soon as', which seems to follow the internal pattern Adverb + *que*, derives from the subordinator in Old Spanish *así como* 'as soon as'. The substitution of *como* 'as' by *que* 'that', the most frequent conjunction in the formation processes of complex conjunctions, was consolidated in the 17th century. The different origin of *así que* 'as soon as' would explain why it did not give rise to the *\*así de que* variant, as well as why it cannot be accompanied by approximative or focal modifiers such as *casi* 'nearly', *justo* 'just', etc., other subordinators formed through the grammaticalization of the pattern Adverb + *que*.

According to Herrero, *así que* 'as soon as' is used in current literary language, although he does not provide examples. I found very limited data in CORPES XXI and PRESSEA. In COSER, I did not document any. Many of these occasional cases illustrate a temporal construction fixed by the use *así que pase X tiempo* "as soon as X time passes", as in (60).

(60)     Ya asoma Dalí, ya suenan sus claros clarines. [Así que pasen-SUBJ.PRES.3PL diez días]
         saldrán a la venta los dos primeros de los ocho tomos de su obra completa. (CORPES
         XXI: "Impune, vivo… y abreviado", *El Cultural*, 21 November 2003, Madrid, Spain)
         'Dalí is already appearing, his clarion calls are already ringing out. As soon as ten days
         go by, the first two of the eight volumes of his complete works will be on sale.'

I have also documented this form with the temporal value of immediate succession in older Catalan speakers with little formal instruction when speaking in Spanish. This usage is produced by interference with the Catalan subordinator *així que* 'as soon as', which has the same meaning, in (61).

(61)   a.   [Así que se le murió-IND.PASTP.3SG la mujer], cerró la zapatería. (J.G., old woman,
             11 November 2022, Romanyà, Pontós, Girona, Spain)
             'As soon as his wife died, he closed the shoe store.'
       b.   [Así que recoja-SUBJ.PRES.1SG la primera aceituna], haré aceite de primera
             prensada. ¿Lo has probado? (B.P., old man, 21 May 2021, Vilamalla, Girona, Spain)
             'As soon as I pick the first olives, I will made first press oil. Have you tried it?'
       c.   [Así que intenta clavarle el cuchillo], él se esconde detrás del burro. (J.E., old man,
             24 March 2002, Vilavenut, Banyoles, Girona, Spain)
             'As soon as he tries to stick the knife in him, he hides behind the donkey.'

### 3.2.6. A la/lo que

The subordinators *a la que* 'as soon as' and *a lo que* 'as soon as' are the results of the grammaticalization of an internal structure that has been discussed by many authors; see Kany (1943; 1945, pp. 435–39), Lope Blanch (1957), and Pavón and Suñer (2018). The most relevant questions are the syntactic and interpretative nature of the forms *la* and *lo* and whether *que* is a conjunction or a relative pronoun. Regarding the first point, as Kany (1943; 1945, pp. 435–39) suggested, the form *la* could be a residue of the elision process of an N with temporal meaning in a structure such *a la (hora) que* 'at the time when'. The category to which *lo* belongs can be deduced by comparing the subordinator *a lo que* 'as soon as' with *en lo que* 'while', which can also be documented from classical Spanish to the present with a different meaning provided by the P (e.g., *En lo que pones la mesa, yo hago la tortilla* 'While you set the table, I make the omelet.'). The meaning of the simultaneity or contingency of *en lo que* 'while' would be constructed by means of the *central coincidence*[3] value of *en* 'in' and the quantifier *lo*, which expresses a segment of time in which the two events coincide. The meaning of the immediate succession of *a lo que* 'as soon as' would instead be obtained with the *terminal coincidence* value of the preposition *a* 'at' and the quantifier *lo*, which, in this case, would indicate a short segment of time corresponding to the interval separating two successive events. Since these subordinators are typical of the colloquial language and have not entered standard Spanish, they do not appear in most diachronic corpora, so it is not possible to reconstruct their evolution with any precision. In any case, the pattern of internal structure they illustrate is not currently productive.

*A la que* 'as soon as' is used in colloquial and rural European Spanish today. In the COSER *corpus*, it is mostly combined with the present indicative, as in (62a), subjunctive, as in (62b), and past imperfect with narrative or historical value, as in (62c), since the interviewees narrated habitual events of their own pasts that could be repeated cyclically. In CORPES XXI, the most common verb form is the past perfect, as in (62d).

(62)  a. [A la que echas-IND.PRES.3SG la canela], se echa una poca de agua. (COSER-1901_01-00, old woman, 30 March 1990, Alboreca, Sigüenza, Guadalajara, Spain)
'As soon as you add the cinnamon, add a little water.'

  b. [A la que lleguen-SUBJ.PRES.1PL a la plaza], suben una cuestica. (COSER-1307_01 old woman, 4 September 2013, Jérica, Castellón, Spain)
'As soon as you get to the square, you get up a little climb.'

  c. [. . .] el arroz [a la que estaba-IND.PAST.IMP.3SG cocido] lo cogían y se lo llevaban a la era. (COSER-1307_01, old woman, 4 September 2013, Jérica, Castellón, Spain)
'Once the rice was cooked, they took it to the threshing floor.'

  d. [A la que afinaron-IND.PASTP.3PL su juego, apretaron su defensa e impusieron su dinamismo ofensivo], se acabó el debate. (CORPES XXI: Ramón Besa, "Juegan todos, todos marcan", www.elpais.com (accessed on 20 July 2023), 2 August 2012, Spain)
'Once they sharpened their play, tightened their defense and imposed their offensive dynamism], the debate was over.'

*A lo que* 'as soon as' is used colloquially in many South American countries, such as Venezuela, Ecuador, Colombia, Chile, Paraguay, Uruguay, and Argentina, except in some regions of the Andes, as in (63).

(63)  a. Mi mama me decía al llegar al trabajo: "Hija, [a lo que fuiste-IND.PASTP.3SG a la Universidad], le levanté el castigo a mi nieto." (*Twitter*: young woman, 17 June 2023, Maracaibo, Venezuela)
'My mother would say to me when I got to work: "Daughter, as soon as you went to university, I took away my grandson's punishment".'

  b. [A lo que entres-SUBJ.PRES.2SG a la casa], ya no hay vuelta atrás. (O.R., young man, Montería, Colombia)
'Once you join the club, there's no turning back.'

  c. ¿Tú vives en Guayaquil???? ¿Caminar con 45 grados y sol asesino? ¿Bicicleta? ¿Que [a lo que sales-IND.PRES.1SG] ya te la robaron? (*Twitter*: young woman, 19 May 2023, Guayaquil, Ecuador)
'Living in Guayaquil? Walking in 45 degrees and killer sun? A bicycle? Stolen as soon as you step outside?'

3.2.7. SQ + que

*Una vez que* 'once' is the result of grammaticalizing the frequentative adjunct *una vez* 'once' together with the generic subordinating conjunction *que* 'that', as in (64); see Eberenz (1982, p. 37) and Espinosa (2010, p. 392).

(64)  a. Se fue en su nave [una vez que acabó-IND.PASTP.3SG su tarea] dejándonos huérfanos. Mayor Tom, te echamos de menos. (*Twitter*: young man, 4 December 2022, Valencia, Spain)
'He took off in his spaceship as soon as he finished his assignment leaving us orphans. Major Tom, we miss you.'

  b. Oliver, que llegó a California en 2006 con una beca Fullbright, planea volver a Lima [una vez acabe-SUBJ.PRES.3SG el doctorado]. (CORPES XXI: www.lostiempos.com (accessed on 4 December 2022), 13 February 2009, Cochabamba, Bolivia)
'Oliver, who came to California in 2006 on a Fullbright Scholarship, plans to return to Lima once he finishes his Ph.D.'

  c. [Una vez que se manifiesta-IND.PRES.3SG la enfermedad], es incurable, solo existen tratamientos que pueden disminuir la sintomatología. (CORPES XXI: Secretaría de Salud, Gobierno de México, www.gob.mex (accessed on 4 December 2022), 15 June 2022, Mexico)
'Once the disease manifests itself, it is incurable; there are only treatments that can alleviate the symptoms.'

According to RAE-ASALE (2009, § 31.14j), its variant *una vez* 'as soon as', as in (65), is characteristic of legal and administrative registers. I followed the diachronic and synchronic trajectory of *una vez que* 'as soon as' and *una vez* 'as soon as' in the available corpora to determine the origin of each form and its geographic and social distribution. The results indicate that *una vez que* 'as soon as' completed its grammaticalization process in the mid-18th century, and *una vez* 'as soon as' appeared in the early 20th century following a parallel path to other complex conjunctions (*tan pronto como > tan pronto; en cuanto que > en cuanto* 'as soon as'). The many examples found on *Twitter* suggest that this form is spreading in the colloquial language of young people.

(65)   a.   Spandau, la cárcel a la que fueron los criminales condenados durante (…) los juicios de Nuremberg, cerró y fue demolida [una vez murió-IND.PASTP.3SG el último condenado R. Hess]. (*Twitter*: young man, 6 March 2023, Limarí, Chile)
'Spandau, the prison to which the convicted criminals went during (…) the Nuremberg trials, was closed and demolished as soon as the last convicted R. Hess died.'

   b.   [Una vez salgas-SUBJ.PRES.2SG de la habitación], no podrás volver. (Twitter: young woman, 8 February 2022, Valencia, Spain)
'Once you leave the room, you will not be able to return.'

   c.   Me gusta ensayar con los actores antes de la filmación pues, [una vez empiezas-IND.PRES.3SG el rodaje], la presión es muy fuerte. (CORPES XXI: www.eladoquintimes.com (accessed on 5 February 2023), 14 May 2018, Puerto Rico)
'I like to rehearse with the actors before shooting because once you start shooting the pressure is very high'.

*Una vez que* 'as soon as' can be realized in some contexts by the variant with the same meaning *una vez de que* 'as soon as', as in (66). This form is mostly documented in COSER (three examples), although two examples appear in CORPES XXI.

(66)   a.   Y [una vez de que las echas-IND.PRES.2SG (las morcillas)], media hora, ties que cogerlas. (COSER-010P, old woman, 26 March 1993, Managarai, Álava, Spain)
'And once you throw them (the black pudding), half an hour, you have to pick them up.'

   b.   Rubalcaba explicó que [una vez de que se produzca-SUBJ.PRES.3SG la impugnación], la decisión final sobre el registro […], es exclusiva del alto tribunal. (CORPES XXI: Paula de las Heras, www.diariodeleon.es (accessed on 5 February 2023), 2 February 2011, León, Spain)
'Rubalcaba explained that once the challenge occurs, the final decision on the registration […], is exclusive to the high court.'

According to the data in COSER (eight examples), and PRESSEA (two examples), the variant with the preposition *de, una vez de* 'as soon as', can govern complements with the verb in a non-personal form in European Spanish, as in (67).

(67)   a.   Y [una vez de hacer-INF la comunión] […] pues a los chicos se les metía en la Congregación de San Luis. (COSER-2004-01, old woman, 6 May 2000, Gabiria, Guipúzcua, Spain)
'And once they had received communion […] the children were sent to the Congregation of St. Louis.'

   b.   [Una vez de cogida-PART la angula], pues se mete en viveros. (COSER-2001-1, old woman, 6 May 2000, Aguinaga, Guipúzcua, Spain)
'Once the elver is harvested, it is put in nurseries.'

   c.   […] porque [una vez de estando-GER juntos] ya no se puede (bailar con otro). (PRESSEA: LPAZ-M22-010, 25 June 2022 La Paz, Bolivia)
'Because once you are together you can no longer (dance with another).'

According to Lenz (1935, p. 203), Kany (1943; 1945, pp. 435–39), and Vidal de Battini (1964, p. 124), temporal subordinates headed by *lo que* (variant of *a lo que*) express

immediate succession in some South American countries. The data collected show that their use is currently limited to the Andean area and that there are variations among different microparameters. The two informants from the North Patagonian area can only use this construction with the present subjunctive, as in (44), while the informant from Ecuador allows both the present subjunctive with prospective value and the simple past with retrospective interpretation, as in (68); see Pavón and Suñer (2018).

(68)    a.    [Lo que {vaya-SUBJ.PRES.1SG ~ *fui-IND.PASTP.1SG} a Chile], {voy ~ *fui-} a comprar materiales de Mapudungun. (M.M. young woman, 15 September 2018, Norpatagonia, Argentina)
            'As soon as I {go ~ *went} to Chile, I will {buy ~ * bought} Mapudungun materials.'

         b.    [Lo que {termine-SUBJ.PRES.1SG ~ *terminé-IND.PASTP.1SG} de dictar la Intro], me {pongo ~ *puse} a trabajar en este artículo. (M.M., young woman, 15 September 2018, Norpatagonia, Argentina)
            'As soon as I {finish ~ *finished} dictating the Intro, I {get ~ *started} to work on this article.'

         c.    [Lo que {vengan-SUBJ.PRES.3PL ~ *vinieron-IND.PASTP.3PL} José y Kerrie], {Podemos ~ *pudimos} ir a casa de Piedra. (middle-aged woman, 9 September 2028, Córdoba, Argentina)
            'As soon as José and Kerrie {come ~ *came}, we {can ~ *were able to} go to Piedra's house.'

         d.    [Lo que {terminemos-SUBJ.PRES.1PL ~ *terminamos-IND.PASTP.1PL} con los médicos], {vamos a ir ~ *fuimos} a verte a Roca. (middle-aged woman, 9 September 2018, Córdoba, Argentina)
            'As soon as we {finish ~ finished} with the doctors, we {will come to see you in Roca.'

         e.    [Lo que Pedro {termine-SUBJ.PRES.3SG ~ *terminó-IND.PASTP.3SG} la casa], {vamos a organizar ~ *organizamos} un asado. (middle-aged woman, 9 September 2018, Córdoba, Argentina)
            'As soon as Pedro {finishes ~ *finished} the house, we {are going to organize ~ *organized} a barbecue.'

(69)    a.    [Lo que vengan-SUBJ.PRES.3PL cosas nuevas al supermercado] iremos-IND.FUT.1PL a verlas. (young woman, 15 October 2028, Ambato, Ecuador)
            'As soon as new things come into the supermarket, we'll go and check them out.'

         b.    [Lo que terminé-IND.PASTP.1SG de trabajar], fui-IND.PASTP.1SG a recoger la compra. (young woman, 15 October 2018, Ambato, Ecuador)
            As soon as I finished work, I went to get the groceries.

From a compositional point of view, the lack of an initial preposition in the subordinator *lo que* 'as soon as' is a clear counterexample to Brucart and Gallego's ([2009] 2016) hypothesis about adverbial subordinates; see Section 1.2. From other theoretical premises, Kany (1943) solved the problem of the absence of P by proposing that there is an embedded P (P "embebida" in Spanish) in this subordinator.

To summarize, we group all the subordinates and their respective variants examined in this section in Table 1.

**Table 1.** Variants of immediate succession subordinators.

| Internal Structure | SUBORDINATOR | Variant 1 | Variant 2 | Variant 3 | Variant 4 |
|---|---|---|---|---|---|
| P + *que* | DESDE QUE | *Desde que* | - | - | - |
| | DE QUE | *De que* | - | - | - |
| Adv + *que* | LUEGO QUE | *Luego que* | *Luego de que* | *Luego de* + INF | - |
| | INMEDIATAMENTE QUE | *Inmediatamente que* | *Inmediatamente de que* | *Inmediatamente de* +INF | - |
| | ENSEGUIDA QUE | *Enseguida que* | *Enseguida de que* | *Enseguida de* +INF | - |

**Table 1.** *Cont.*

| Internal Structure | SUBORDINATOR | Variant 1 | Variant 2 | Variant 3 | Variant 4 |
|---|---|---|---|---|---|
| Adv + *que* 2 | LUEGUITO QUE | *Lueguito que* | *Lueguito de que* | *Lueguito de* + INF | - |
| | ENANTITO QUE | *Enantito que* | *Enantito de que* | *Enantito de +*INF | - |
| | LUEGO LUEGO QUE | *Luego luego que* | *Luego luego de que* | *Luego luego de +*INF | - |
| Adv + V-FINITE | APENAS | *Apenas* | *Apenas* + PART | *Apenas que* | *Apenas* + INF |
| | NO BIEN | *No bien* | *No bien* + PART | *No bien que* | *No bien* + INF |
| | NI BIEN | *Ni bien* | *Ni bien* + PART | *Ni bien que* | *Ni bien* + INF |
| | RECIÉN | *Recién* | *Recien* + PART | - | - |
| Adv + INF | NADA MÁS | *Nada más +*INF | *Nada más de* + INF | *Nada más que* | *Nada más* + VFINITE |
| | NOMÁS | *Nomás +*INF | *Nomás de* + INF | *Nomás que* | *Nomás* + VFINITE |
| Modal + SQ structures | TAN PRONTO COMO | *Tan pronto como* | *Tan pronto* + VFINITE | *Tan pronto* + INF | - |
| | TAN LUEGO COMO | *Tan luego como* | *Tan luego* + VFINITE | - | - |
| | ASÍ QUE | *Así que* | - | - | - |
| | EN CUANTO | *En cuanto* | *En cuanto que* | *Cuanto que* | - |
| A +*la/lo que* | A LA QUE | *A la que* | - | - | - |
| | A LO QUE | *A lo que* | - | - | - |
| SQ (+ N) + *que* | UNA VEZ QUE | *Una vez que* | *Una vez* + VFINITE | *Una vez de que* | *Una vez de* + PART |
| | LO QUE | *Lo que* | - | - | - |
| Hybrid constructions | AL NOMÁS | *Al nomás+* INF | *Al nomás que* | - | - |
| | AL NADA MÁS | *Al nada más* + INF | - | - | - |
| | AL APENAS | *Al apenas+* INF | - | - | - |

## 4. Discussion

In this paper, we have analyzed the variation in immediate succession subordinates in current Spanish from a compositional perspective. According to this theoretical framework, the meaning of immediate succession is not embodied exclusively in the temporal subordinator but is divided into different relevant components of the sentence. In other words, the meaning of immediate succession between two events is constructed compositionally by means of the subordinator heading the temporal subordinate, together with the tense, the mood, the syntactic aspect of the verbs involved, their lexical aspect (*Aktionsart*), and, optionally, the presence of temporal and aspectual modifiers, quantification, negation, and the relative order between the main sentence and the subordinate. The article has focused on subordinators and the variants in which they can materialize in different contexts.

From the data analysis carried out in Section 3 and summarized in Table 1, we can draw several generalizations.

First, the current system of subordinators of immediate succession is characterized by the presence of various patterns of formation of its internal structure. Some of these patterns (P + *que*, modal and SQ structures, *a la/lo que*, and SQ(N) + *que*) have long ceased to be productive although some subordinators formed by means of these patterns have remained as fossils in the current language. It is also confirmed that the productive patterns for the formation of immediate succession subordinators in the current language are Adv + QUE/+INF/+PART and Adv + VFINITE. All of them conform to the initial rule for the formation of adverbial subordinators, P + que (cf. Section 1.2), since the superficial divergences they possess are due to the different forms that the verb of the subordinate can take.

Second, it can also be observed that different variants of the same subordinator are obtained by applying general grammatical rules. Their function is usually to extend the range of complements that can be governed by the subordinator. For example, the recently created subordinators *nada más* 'as soon as' and y *nomás 'as soon as'* head an infinitive

sentence in their first occurrences, but later, the variants *nada más que* 'as soon as' and *nomás que* 'as soon as' govern a subordinate clause with a finite verb. Other subordinators, such as *apenas* 'as soon as' and *no bien* 'as soon as' (and *recién* 'as soon as', whose process of grammaticalization is in progress), can introduce finite verb subordinates without merging with the generic conjunction *que* '*that*'. In these cases, the subordinator has a dual function, equivalent to relatives in a free relative subordinate clause: on the one hand, providing the sense of immediacy, and on the other hand, marking the beginning of the subordinate clause they govern. The subordinators *apenas* 'as soon as' and *recién* 'as soon as' coincide in the current language with homophonic forms that are in an intermediate period of grammaticalization, such as the structures of inverse *cuando* (which originated in Medieval Spanish), and, in the present, especially in the colloquial Spanish registers, with correlations with the coordinating conjunction *y* 'and' (*apenas/no bien . . . y*). In the latter construction, it is not clear how the meaning of immediacy between the two events is constructed, so it would be desirable to analyze it from a formal point of view to determine what the limits between subordination structures and this type of correlatives are.

Third, the analysis of the data has also shown that the subordinates that fit the productive pattern have numerous variants and, in addition, can hybridize with other temporal subordinator patterns. This is the case for the subordinators in consolidation *al nomás* + INF > *al nomás que* '+ VFINITE 'as soon as' and *al nada más* + INF 'as soon as'. The fact that the subordinator *al nada más* 'as soon as' does not have the variant *al nada más que* + VFINITE would indicate that the evolutionary process is faster in areas such as Mexico, where *al nomás* and *al nomás que* have been formed, than in European Spanish.

Fourth, in the formation of variants, we observe processes that seem to go in opposite directions. On the one hand, there are well-consolidated subordinators such as *apenas*, *no bien,* and *ni bien* 'as soon as', which currently have variants fused with the generic conjunction *que* (in very marginal colloquial registers of American Spanish, *apenas que, no bien que*, and *ni bien que* 'as soon as', respectively). On the other hand, subordinators such as *tan pronto como* 'as soon as' and *una vez que* 'as soon as' have evolved into variants without the conjunction.

Finally, newly subordinators that have been created by means of the grammatical processes discussed in this article may have an ephemeral life and thus fade from the language without a trace, but if they constitute a relevant cue for a new generation of speakers (see Kroch 2001, 2005), they may initiate a process of consolidation and become part of the language of the future.

**Funding:** This research was funded by MCIUC (Ministeri de Ciència, Innovació i Universitats), grant number PID 2021-123617-NB-C42, Spain.

**Institutional Review Board Statement:** Not applicable.

**Informed Consent Statement:** Informed consent was obtained from all subjects involved in the study.

**Data Availability Statement:** All of the data contained in this study has been obtained by the author from the corpora listed in Section 2.

**Acknowledgments:** I am grateful to all the informants who have generously provided me with data on their variety of Spanish: Álvaro Cruz, Ángela Di Tullio, Joan Estradé, Facundo González, Joaquima Gratacós, María Mare, Alicia Martin, Baldiri Palomeras, and Luis Valle. Needless to say, all the usual disclaimers apply.

**Conflicts of Interest:** The author declares no conflict of interest.

## Notes

[1]   Grammaticalization is a diachronic process by which a lexical item takes on grammatical functions. It is also called the process by which a grammatical item develops a new grammatical value; see Elvira (2015, p. 93). (Lehman 1985) was the first to establish that grammaticalization processes are made up of a series of relatively independent subprocesses of change that affect three quantitative or qualitative parameters of the form or grammatical function of the expressions being grammaticalized. These three parameters are weight, cohesion, and variability, which affect different aspects of the paradigmatic and syntagmatic

behaviour of the grammaticalized items. From a paradigmatic point of view, grammaticalization may involve subprocesses such as the phonetic reduction, paradigmatization, and obligatoriness of the grammaticalized item, while on a syntagmatic level, the subprocesses involve the condensation, coalescence, and fixation of the order of the grammaticalized item. The decomposition into different independent subprocesses makes it possible to explain, more precisely, the fact that some of the phonetic, syntactic, and semantic changes involved in grammaticalization do not occur simultaneously in all grammaticalized expressions; see Hopper and Traugott ([2003] 2012). The main objective of this chapter is to describe the geographical variation of current Spanish; therefore, I will refer only to the grammaticalization processes of some subordinators when it is useful to understand the data on the current language. For a detailed analysis of the properties of grammaticalized complex conjunctions, see Pavón (1999, § 9.5; 2012, § 3.2 and § 6.2).

2   In Medieval and Classical Spanish, *(en) quanto (que)* 'while' was a subordinator of simultaneity since the relative quantifier *quanto* 'how much' referred to a portion of time in which the two events of a complex period coincided. According to Herrero (2005, p. 242; 2018), the meaning of immediate succession appears at the end of the 18th century, when the subordinator was fully grammaticalized. In this second meaning, the quantifier *cuanto* 'how much' does not express the duration in which the two events coincide, but the short lapse of time that measures the interval between them. Ridruejo (2003, pp. 238–30) argues that such a change is produced by a process of exaptation, since an already grammaticalized element acquires a new interpretation in certain contexts in which two punctual events are related, and subsequently, this change extends to the whole paradigm.

3   Spatial relations have been classified following the distinction between *terminal coincidence* and *central coincidence*, defined in Hale (1985, pp. 239–40). Ps of *central coincidence* such as *en* 'in' relate two entities in a constant, static, and unchanging way, while Ps of *terminal coincidence* relate them in a dynamic way.

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
