# Peer review of "Levels of Variation in Subordinates of Immediate Succession in Current Spanish"

_languages, doi:10.3390/languages9010016_

Round 1
Reviewer 1 Report
Comments and Suggestions for Authors
Please see attached file.

Author Response
Please see the atachment

Reviewer 2 Report
Comments and Suggestions for Authors
See attached file

The English in this paper is very good. However, there are some areas in which it is clear that the writer is not a native speaker, as some sentences are ungrammatical. I recommend having the paper proofread by a native English speaker before publication.
Author Response
Please, see the attachment

Round 2
Reviewer 1 Report
Comments and Suggestions for Authors
The author has carefully addressed my comments as well as the suggestions of the other reviewer. The paper is much better now. However, there are still some points that could be improved. First, in section 1.2 the analysis of the contribution of tenses is rather superficial and could be improved. In terms of terminology, I would use "imperfective past" for the Imperfecto (instead of "past imperfect", as the author does, line 244). Also, the author says of the "pretérito anterior" that the perfective value is hyperspecified (line 250). Why? And is this the case for other perfective forms with the auxiliary "haber" and a past participle?
Second, I find that the goal of providing an overview of "levels of variation" in this domain is too ambitious and leads to a certain superficiality in the analysis. The author focuses mostly on regional variation of the subordinators as well as diachronic variation. Why not say that, rather than announcing an aim in the introduction that does not really correspond to what is actually done in the paper?
Comments on the Quality of English LanguageI'm not a native speaker of English, but the language looks fine in general.
Author Response
Please, see the attachment
